# Peptidoglycan-Chi3l1 interaction shapes gut microbiota in intestinal mucus layer

**Yan Chen[†], Ruizhi Yang[†], Bin Qi\*, Zhao Shan\***

Southwest United Graduate School, Yunnan Key Laboratory of Cell Metabolism and Diseases, State Key Laboratory of Conservation and Utilization of Bio-resources in Yunnan, Center for Life Sciences, School of Life Sciences, Yunnan University, Kunming, China

**Abstract** The balanced gut microbiota in intestinal mucus layer plays an instrumental role in the health of the host. However, the mechanisms by which the host regulates microbial communities in the mucus layer remain largely unknown. Here, we discovered that the host regulates bacterial colonization in the gut mucus layer by producing a protein called Chitinase 3-like protein 1 (Chi3l1). Intestinal epithelial cells are stimulated by the gut microbiota to express Chi3l1. Once expressed, Chi3l1 is secreted into the mucus layer where it interacts with the gut microbiota, specifically through a component of bacterial cell walls called peptidoglycan. This interaction between Chi3l1 and bacteria is beneficial for the colonization of bacteria in the mucus, particularly for Gram-positive bacteria like *Lactobacillus*. Moreover, a deficiency of Chi3l1 leads to an imbalance in the gut microbiota, which exacerbates colitis induced by dextran sodium sulfate. By performing fecal microbiota transplantation from Villin-cre mice or replenishing *Lactobacillus* in IEC$^{\Delta Chil1}$ mice, we were able to restore their colitis to the same level as that of Villin-cre mice. In summary, this study shows a 'scaffold model' for microbiota homeostasis by interaction between intestinal Chi3l1 and bacteria cell wall interaction, and it also highlights that an unbalanced gut microbiota in the intestinal mucus contributes to the development of colitis.

**\*For correspondence:**
qb@ynu.edu.cn (BQ);
shanzhaolab@163.com (ZS)

[†]These authors contributed equally to this work

**Competing interest:** The authors declare that no competing interests exist.

## eLife assessment

Supported by **convincing** data, this **valuable** study demonstrates that the Chitinase 3-like protein 1 (Chi3l1) interacts with gut microbiota and protects animals from intestinal injury in laboratory colitis model. The revised article sufficiently addressed the reviewers' comments. The work will be of interest to scientists studying crosstalk between gut microbiota and inflammatory diseases.

## Introduction

Intestinal homeostasis is crucial for maintaining human health (*Lozupone et al., 2012*). Alterations in gut microbiota composition have been linked to various diseases including cancer, obesity, and neurological disorders (*Bäumler and Sperandio, 2016*; *Charbonneau et al., 2016*; *Honda and Littman, 2016*; *Sonnenburg and Bäckhed, 2016*; *Thaiss et al., 2016*). Dysbiosis, which refers to an imbalance in gut microbiota, is characterized by decreased microbial diversity, the presence of harmful microbes, or absence of beneficial ones (*Petersen and Round, 2014*). Therefore, understanding the factors that influence gut microbiota is a fundamental goal in microbiome research (*McDonald et al., 2020*). Growing evidence suggests that colonization of the gut mucus layer can affect the susceptibility and progression of intestinal diseases like inflammatory bowel disease (IBD), irritable bowel syndrome, and celiac disease (*Gordon, 2012*). IBDs, such as Crohn's disease (CD) and ulcerative colitis (UC), are characterized by chronic inflammation of the intestinal mucosa. Although the cause of the IBD is unclear,

mouse models lacking the key components of the mucus are predisposed to colitis, accompanied by dysbiosis in mucosa (*Fu et al., 2011*; *Johansson et al., 2008*), which is in accordance with increased epithelial-adherent microbial communities in biopsies from patients with IBD (*Johansson et al., 2014*; *Swidsinski et al., 2005*). Furthermore, there were significant differences in the gut microbiota of CD patients compared with healthy controls, and these differences were only present in mucus samples (not stool samples), suggesting that bacteria in the mucus layer may be more important for the development of IBD (*Gevers et al., 2014*). Donaldson et al. discovered that the intestinal flora utilizes host immunoglobulin A (IgA) for mucus colonization, indicating that the host may secrete certain factors to maintain intestinal flora homeostasis in the mucus (*Donaldson et al., 2018*). However, the mechanisms regulating gut microbiota colonization in the mucus layer remain largely unknown. We aim to investigate the regulation of microbial communities in gut mucus and its implications in intestinal diseases.

Chitinase 3-like protein 1 (Chi3l1, also known as YKL-40 in humans) is a secreted protein that belongs to the glycosylhydrolase 18 family (*Lee et al., 2011*). Despite its name, Chi3l1 can bind to chitin but does not have chitinase activity (*Houston et al., 2003*). In our investigation, we noticed that chitin and peptidoglycan (PGN), a major component of bacterial cell walls, have similar structures (*Fulde et al., 2018*). Based on this information, we speculate that Chi3l1 might also interact with PGN and, therefore, interact with bacteria. Interestingly, Chi3l1 is expressed in intestinal epithelial cells (IECs) and the lamina propria. We hypothesize that Chi3l1 may be secreted by IECs and regulate the gut microbiota through its interaction with PGN in the mucus layer. In our study, we discovered that gut microbiota induced the expression of Chi3l1 in epithelial cells. Once expressed, Chi3l1 is secreted into the mucosa and interacts with bacteria, particularly with the bacterial cell wall component PGN. This interaction promotes bacterial colonization, especially of beneficial bacteria such as *Lactobacillus*. As a result, mice with higher levels of Chi3l1 are less susceptible to colitis.

## Results

### Intestinal epithelial cells express Chi3l1 induced by gut microbiota

The gut microbiota's composition is shaped by host factors, including IgA (*Donaldson et al., 2018*), RegIIIγ (*Vaishnava et al., 2011*), and TLR-5 (*Fulde et al., 2018*). Yet, specific host factors which maintain the homeostasis of the microbiota remain largely undefined. Drawing on the theory of co-evolution between the host and microbiota (*Groussin et al., 2020*), we propose that host factors, which are induced by bacteria in the gut, could play a pivotal role in regulating bacterial colonization and composition. In a previous study, it was observed that both live *Escherichia coli* and heat-killed *E. coli* treatment resulted in a significant increase in the expression of the gene encoding Chi3l1 in human intestinal organoids (*Hill et al., 2017*). To verify this finding, we conducted immunohistochemical staining on intestinal tissue sections of germ-free and specific pathogen free (SPF) C57BL/6J wildtype mice. We observed a substantial increase of Chi3l1 expression in SPF mice compared to germ-free mice (*Figure 1A*). The intestinal epithelium comprises various cell types, including intestinal cells, goblet cells, endoentercrine cells, Tuft cells, Paneth cells, M cells, and others (*Johansson et al., 2014*). To identify the cellular sources of Chi3l1, we performed co-staining with markers for specific cell types, including chromogranin A (ChgA) for enteroendocrine cells, *Ulex europaeus* Agglutinin I (UEA-1) for goblet cells and Paneth cells, and double cortin-like kinase 1 (DCLK1) for tuft cells. Our results revealed that Chi3l1 is primarily expressed in enteroendocrine cells in the ileum and goblet or Paneth cells in the colon (*Figure 1B*). However, Chi3l1 expression was not observed in tuft cells (*Figure 1—figure supplement 1A*).

Furthermore, we isolated total bacteria from wildtype mouse feces and treated DLD-1 cells (a colorectal adenocarcinoma cell line) with the bacterial extract for 12 hr. We found that the bacterial extract directly induced Chi3l1 expression in DLD-1 cells (*Figure 1C*). To examine whether the induction of Chi3l1 expression requires a specific bacteria strain, we further identified the bacterial extract using 16S rRNA sequencing. Our results revealed that *E. coli* specifically stimulated Chi3l1 expression in DLD-1 cells, while *Staphylococcus sciuri* did not have the same effect (*Figure 1D*). Although our data are limited to these two bacterial strains, it suggests that not all bacteria can induce the expression of Chi3l1. Next, we wondered what component of bacteria can induce Chi3l1 expression. We tried heat-killed *E. coli*, which maintains bacterial cell wall integrity. We found that treatment of DLD-1

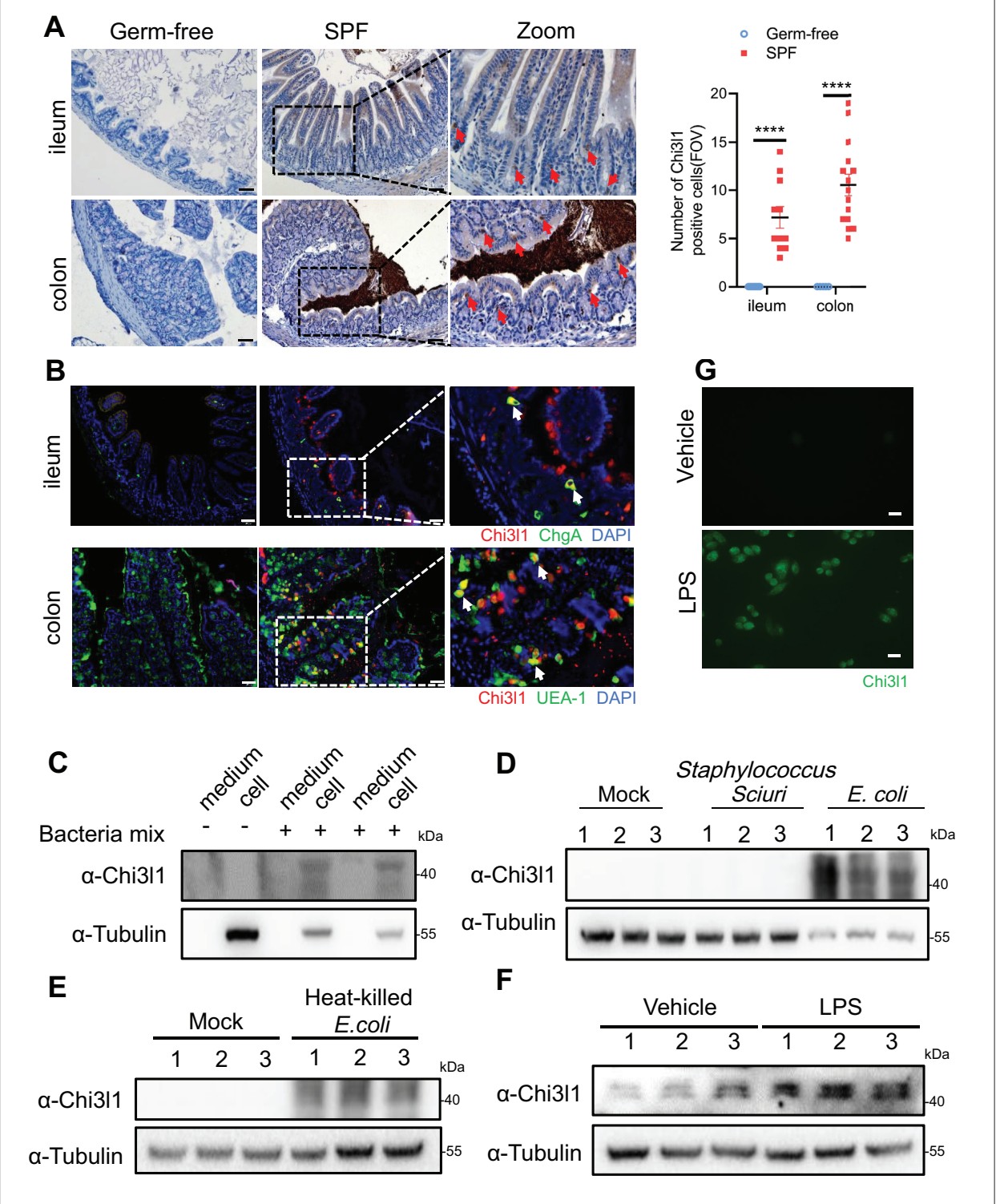

**Figure 1.** Intestinal epithelial cells express Chi3l1 induced by gut microbiota. (**A**) Immunohistochemical (IHC) staining to detect Chi3l1 in both ileum and colon from germ-free and wildtype mice. Ctrl (wildtype mice without application of first antibody), WT (wildtype C57B/6J mice). Red arrows indicate Chi3l1-expressing cells. Scale bars, 50 μm (Ctrl, Germ-free, WT). The number of Chi3l1-positive cells in each field of view (FOV) was analyzed. (**B**) Ileum and colon were collected from wildtype mice and stained with ChgA (green), Chi3l1(red), and nuclear DAPI (blue) in ileum and UEA-1 (green), Chi3l1 (red), and nuclear DAPI (blue) in colon. Scale bars, 50 μm. Ctrl (without application of first antibody), WT (wildtype C57BL/6J mice). (**C**) Western blot to detect Chi3l1 protein expression in DLD-1 cells after bacteria mix infection for 12 hr. Bacteria mix are total bacteria extracted from feces of wildtype mice. (**D**) Western blot to detect Chi3l1 protein expression in DLD-1 cells after *Staphylococcus sciuri* and *E. coli* infection for 12 hr. *Staphylococcus sciuri*

*Figure 1 continued on next page*

*Figure 1 continued*

and *E. coli* are isolated from bacteria mix and verified by 16S rRNA sequencing. Three independent experimental results are showed. (**E**) Western blot to detect Chi3l1 protein expression in DLD-1 cells after treatment with heat-killed *E. coli* for 12 hr. Three independent experimental results are showed. (**F**) Western blot to detect Chi3l1 protein expression in DLD-1 cells after treatment with 100 pg/mL lipopolysaccharides (LPS) for 12 hr. Three independent experimental results are showed. (**G**) Immunofluorescence to detect Chi3l1 protein expression in DLD-1 cells after treatment with 100 pg/mL LPS for 12 hr. Scale bars, 20 µm. The presence of cells in the untreated sample is annotated using white dashed lines based on the overexposure. All data above represent at least three independent experiments. Representative images are shown in (**A, B**), n = 3 mice/group. *p<0.05, **p<0.01, ***p<0.001, ****p<0.0001, ns: no significant difference, error bar indicates SEM.

The online version of this article includes the following source data and figure supplement(s) for figure 1:

**Source data 1.** File containing original western blots for *Figure 1C–F*, indicating the relevant bands.

**Source data 2.** Original files for western blot analysis displayed in *Figure 1C–F*.

**Source data 3.** Numerical data of *Figure 1A*.

**Figure supplement 1.** Chi3l1 do not express in tuft cells.

**Figure supplement 1—source data 1.** File containing original DNA gels for *Figure 1—figure supplement 1B*, indicating the relevant bands.

**Figure supplement 1—source data 2.** Original files for DNA gels for *Figure 1—figure supplement 1B*.

cells with heat-killed *E. coli* also led to an induction of Chi3l1 expression (*Figure 1E*). These findings suggest that the induction of Chi3l1 expression does not necessarily require live bacteria and that bacterial components alone are sufficient to induce this response.

Given that *E. coli* is Gram-negative and *S. sciuri* is Gram-positive, we hypothesized that the difference in their ability to induce Chi3l1 expression might be due to variations between Gram-negative and Gram-positive bacteria, such as the presence of lipopolysaccharides (LPS). To test this hypothesis, we used LPS to induce Chi3l1 expression. Consistent with our hypothesis, LPS successfully induced Chi3l1 expression (*Figure 1F and G*). Collectively, these findings provide evidence that the gut microbiota can induce Chi3l1 expression in IECs. Collectively, these findings provide evidence that the gut microbiota can induce Chi3l1 expression in IECs.

## Chi3l1 interact with bacteria via peptidoglycan

Chi3l1 belongs to a group of proteins called non-enzymatic chitinase-like proteins, which are known to bind chitin. Chitin is a polysaccharide present in the exoskeleton of arthropods and the cell walls of fungi (*Zhao et al., 2020*). By comparing the structure of chitin with that of PGN, a component of bacterial cell walls, we observed that they have similar structures (*Figure 2A*). Based on this observation, we hypothesized that Chi3l1 may interact with gut bacteria through PGN. To test our hypothesis, we conducted co-incubation experiments where we mixed recombinant mouse Chi3l1 (rmChi3l1) with either Gram-positive or Gram-negative bacteria and then precipitated the bacteria through centrifugation. We found that rmChi3l1 was present in the pellet obtained from both Gram-positive and Gram-negative bacteria (*Figure 2B*), suggesting that Chi3l1 can directly interact with bacteria.

To further investigate the interaction between Chi3l1 and PGN, we also co-incubated PGN with rmChi3l1 and precipitated the PGN through centrifugation. PGN is an insoluble substance and hence can be precipitated by centrifugation. Consistent with our previous results, we observed that rmChi3l1 was present in the pellet obtained from PGN but not in the pellet obtained from bovine serum albumin (BSA), which served as a negative control (*Figure 2C*). Furthermore, we also examined the interaction between PGN and recombinant human Chi3l1 (rhChi3l1) and obtained similar results (*Figure 2D*). These findings indicate that Chi3l1 interacts with bacteria through PGN.

To better characterize the binding between Chi3l1 and PGN, we compared the binding affinities of Chi3l1 to both PGN and chitin. We incubated chitin and PGN with rmChi3l1 in increasing doses (25, 50, 100 µg) and detected the precipitated rmChi3l1 by either chitin or PGN. Our results indicate that Chi3l1 interacts with PGN in a dose-dependent manner (*Figure 2E*). In contrast, the binding between Chi3l1 and chitin did not exhibit dose dependency (*Figure 2E*). These findings suggest a specific and distinct binding mechanism for Chi3l1 with PGN compared to chitin.

To investigate whether the Chi3l1-PGN interaction could facilitate the colonization of Gram-positive bacteria, we conducted adhesion experiments using DLD-1 cells and bacteria. We employed a GlmM mutant (PGN synthesis-deficient) and K12 bacteria (a wildtype *E. coli* strain used as a control) to assess their adhesion capabilities. The results showed that the adhesion ability of the GlmM mutant

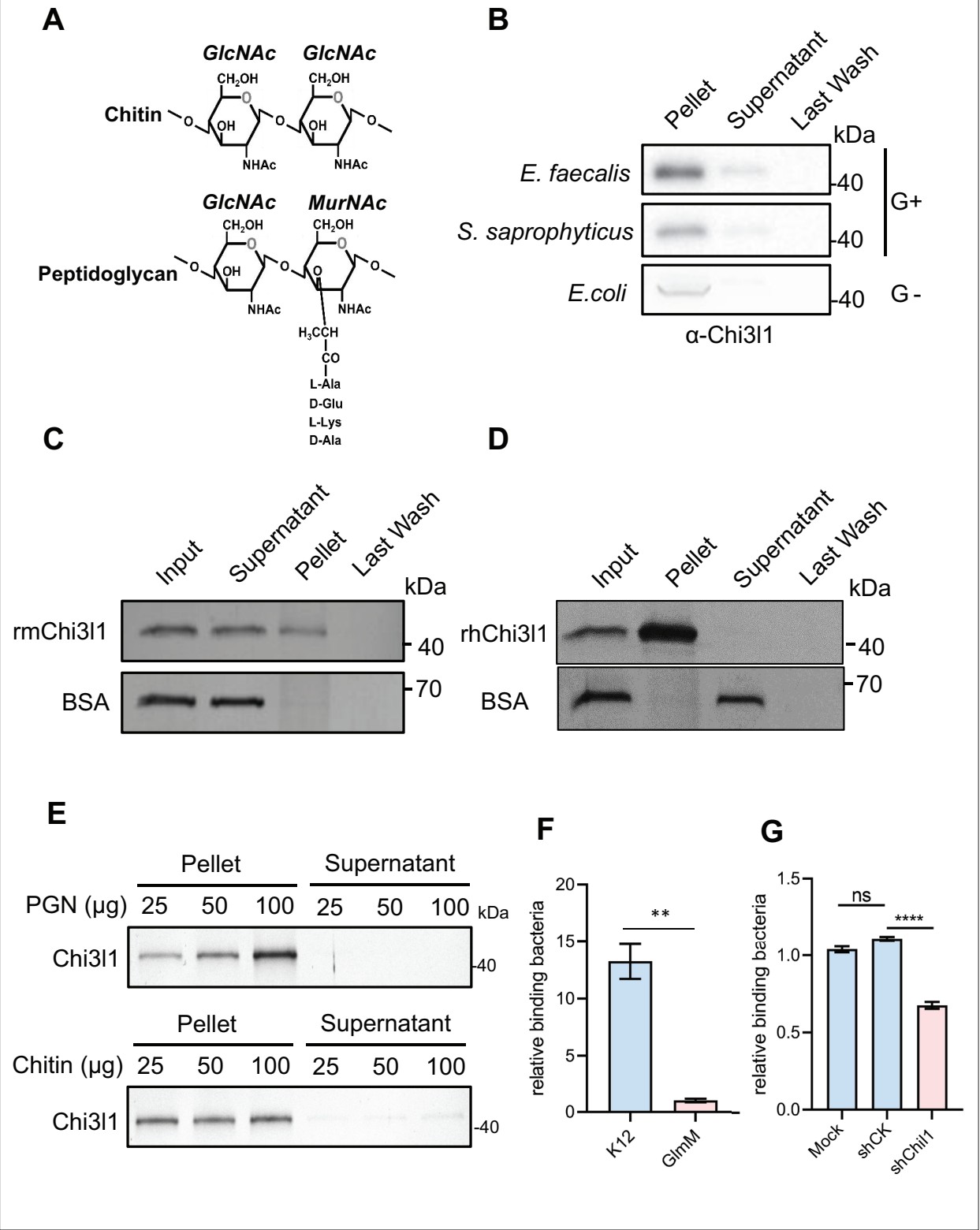

**Figure 2.** Chi3l1 interact with bacteria via peptidoglycan (PGN). (**A**) Structural comparison between chitin and PGN. Both chitin and PGN contain N-acetylglucosamine (GlcNAc) and have β–1,4-glycosidic bonds in their structures. However, chitin is purely a polysaccharide, while PGN includes a peptide component that forms cross-links between chains (*Zhou et al., 2022*). (**B**) Gram-positive bacteria (*E. faecalis*, *S. saprophyticus*) and Gram-negative bacteria (*E. coli*) were incubated with 1 μg of recombinant mouse Chi3l1 protein (rmChi3l1), respectively. Proteins bound to indicated bacteria were precipitated by centrifugation. Western blot was used to detect rmChi3l1 in Pellet, Supernatant (unbound proteins) and Last Wash (last wash

*Figure 2 continued on next page*

*Figure 2 continued*

unbound proteins). (**C**) Insoluble PGN were incubated with either recombinant mouse Chi3l1 protein (rmChi3l1) or bovine serum albumin (BSA). Proteins bound to PGN were precipitated by centrifugation. Silver staining was used to detect rmChi3l1 in Input, Supernatant (unbound proteins), Pellet and Last Wash (last wash unbound proteins). (**D**) Insoluble PGN were incubated with recombinant human Chi3l1 protein (rhChi3l1). Proteins bound to PGN were precipitated by centrifugation. Silver staining was used to detect rhChi3l1 in Input, Supernatant (unbound proteins), Pellet and Last Wash (last wash unbound proteins). All data above represent at least three independent experiments. (**E**) Insoluble PGN or chitin was incubated with rmChi3l1. Chi3l1 bound to PGN (upper panel) and chitin (lower panel) was precipitated and detected by silver staining. The supernatant represents the last wash, and the pellet contains proteins precipitated by either PGN or chitin. (**F**) Relative DLD-1 bacterial binding preference after treatment with K12 or GlmM, a PGN synthesis-deficient mutant. Colony-forming units (CFU) were counted, and GlmM CFU were normalized to 1. (**G**) Relative K12 bacterial adhesion preference after DLD-1 cells were transfected without (Mock), or with scramble shRNA (shCK), or with sh*Chil1*. CFU were counted, and the Mock group were normalized to 1. *p<0.05, **p<0.01, ***p<0.001, ****p<0.0001, ns: no significant difference, error bar indicates SEM.

The online version of this article includes the following source data for figure 2:

**Source data 1.** File containing original western blots for *Figure 2B* and silver staining for *Figure 2C–E*, indicating the relevant bands.

**Source data 2.** Original files for western blot analysis displayed in *Figure 2B* and silver staining for *Figure 2C–E*.

**Source data 3.** Numerical data of *Figure 2F and G*.

to cells significantly decreased (*Figure 2F*). Additionally, after knocking down Chi3l1 in DLD-1 cells (knockdown efficiency over 50%), we observed a decrease in bacterial adhesion (*Figure 2G*). These findings suggest that the Chi3l1-PGN interaction plays a crucial role in bacterial adhesion.

## Intestinal bacteria are disordered in IEC$^{\Delta Chil1}$ mice, especially Gram-positive bacteria

To gain initial insights into how the expression of Chi3l1 in IECs affects the colonization of gut microbiota, we created mice with a specific deficiency of Chi3l1 in IECs (referred to as IEC$^{\Delta Chil1}$ mice) (*Figure 3—figure supplement 1*). We then conducted bacterial 16S rRNA sequencing analysis of the colon contents of both Villin-cre and IEC$^{\Delta Chil1}$ littermates. Our analysis of alpha diversity revealed that the bacterial population was relatively lower in IEC$^{\Delta Chil1}$ littermates compared to Villin-cre littermates (*Figure 3A*). This finding was further confirmed by conducting universal bacterial 16S rRNA qPCR analysis of the feces and ileum contents of IEC$^{\Delta Chil1}$ and Villin-cre littermates, which also showed lower bacterial enrichment in IEC$^{\Delta Chil1}$ mice (*Figure 3B*). Furthermore, principal component analysis demonstrated significant differences in bacterial diversity between Villin-cre and IEC$^{\Delta Chil1}$ littermates (*Figure 3C*).

When we examined the relative abundance of Gram-positive and Gram-negative bacteria between Villin-cre and IEC$^{\Delta Chil1}$ littermates, we observed that Gram-positive bacteria were significantly reduced in IEC$^{\Delta Chil1}$ mice, while there was no notable difference in Gram-negative bacteria (*Figure 3D*). This result was further validated by staining lipoteichoic acid (LTA), a component present on Gram-positive bacteria, which revealed a lower abundance of Gram-positive bacteria in IEC$^{\Delta Chil1}$ compared to Villin-cre littermates (*Figure 3E*). Moreover, visualization of *Firmicutes* by bacteria fluorescence in situ hybridization (FISH) staining, a dominant group of Gram-positive bacteria in the gut, also showed reduced levels of *Firmicutes* in the colon lumen of IEC$^{\Delta Chil1}$ mice compared to Villin-cre mice (*Figure 3F*). Analysis of the relative abundance of specific Gram-positive bacterial species demonstrated a significant reduction in *Lactobacillus* in IEC$^{\Delta Chil1}$ mice compared to Villin-cre mice (*Figure 3G*). Similar results were observed in *Chil1$^{-/-}$* mice compared to wildtype mice (*Figure 3H*). Consist with the 16S rRNA sequencing analysis data, qPCR results showed that *Turicibacter* was more abundant in IEC$^{\Delta Chil1}$ mice than Villin-cre mice (*Figure 3—figure supplement 2*). These findings suggest that Chi3l1 plays a role in regulating the colonization of Gram-positive bacteria, particularly *Lactobacillus*, in the murine gut.

## Chi3l1 promotes the colonization of Gram-positive bacteria in intestinal mucus

Chi3l1 was found in secretory cells like goblet cells and Paneth cells, suggesting that it may be secreted into the intestinal lumen (*Figure 1B*). Immunohistochemistry staining of Chi3l1 in the colon revealed a large amount of Chi3l1 signals in the mucus layer (*Figure 4A*). Immunofluorescence co-staining of Chi3l1 with UEA-1 in the colon yielded similar results (*Figure 4B*). Furthermore, Chi3l1 was also detected in the ileum and colonic tissues and mucus layer (*Figure 4C and D*). These findings indicate

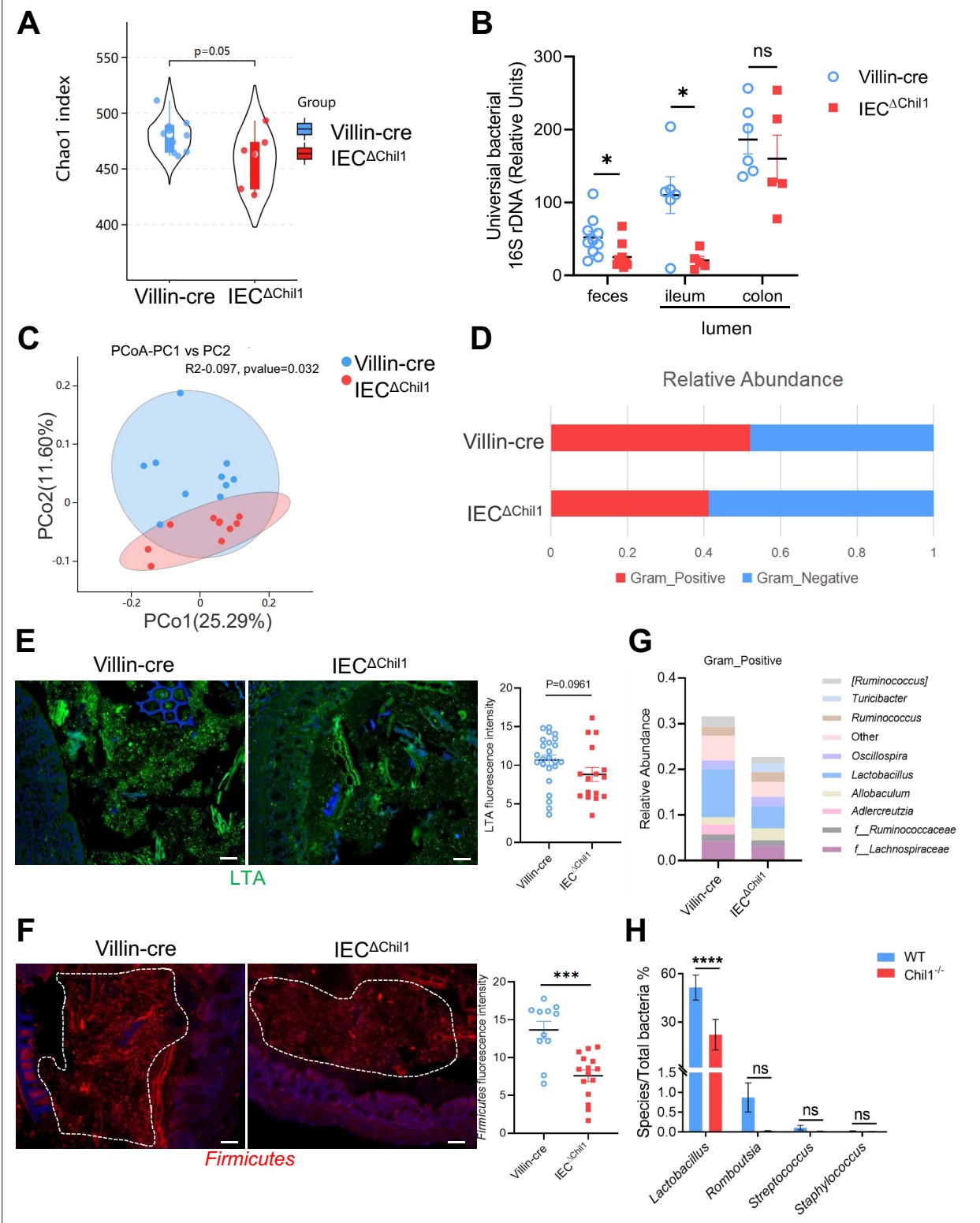

**Figure 3.** Intestinal bacteria are disordered in IEC$^{\Delta Chil1}$ mice, especially Gram-positive bacteria. (**A, C, D, G**) Female Villin-cre and IEC$^{\Delta Chil1}$ littermates continue to cage together after weaning for 8 weeks. Microbial communities in feces and intestinal lumen were characterized by 16S rRNA sequencing. n = 7 or 10/group. (**A**) Alpha diversity analysis of colon contents between Villin-cre and IEC$^{\Delta Chil1}$ littermates. (**B**) qPCR analysis of total bacteria in the feces and ileum, colon luminal microbial communities of Villin-cre and IEC$^{\Delta Chil1}$ littermates. Values for each bacterial group are expressed relative to total 16S rRNA levels. n = 5–10/group. (**C**) Principal component analysis of weighed UniFrac distances of 16S community profiles of Villin-cre and IEC$^{\Delta Chil1}$

*Figure 3 continued on next page*

*Figure 3 continued*

littermates feces (binary-jaccard). (**D**) Relative abundance of Gram-positive and Gram-negative bacteria in colon contents of Villin-cre and IEC^ΔChil1^ littermates are shown. (**E**) Lipoteichoic acid (LTA) (green) was detected by immunofluorescence in colon sections of Villin-cre and IEC^ΔChil1^ littermates. Nuclei were detected with DAPI. Scale bars, 50 μm. The average fluorescence intensity in each field of view (FOV) was analyzed. (**F**) Fluorescence in situ hybridization (FISH) detection of Gram-positive bacteria (red) in the colon of Villin-cre and IEC^ΔChil1^ littermates, nuclei were detected with DAPI (blue). Scale bars, 50 μm. The average fluorescence intensity in each FOV was analyzed. (**G**) Relative abundance of Gram-positive bacteria genera in colon lumen of Villin-cre and IEC^ΔChil1^ littermates. (**H**) Female wildtype and *Chil1^-/-^* littermates continue to cage together after weaning for 8 weeks. Microbial communities in feces were characterized by 16S rRNA sequencing. n = 3 mice/group. Representative images are shown in (**E, F**), n = 4–5/3-6 mice/group. *p<0.05, **p<0.01, ***p<0.001, ****p<0.0001, ns: no significant difference, error bar indicates SEM.

The online version of this article includes the following source data and figure supplement(s) for figure 3:

**Source data 1.** Numerical data of *Figure 3A–H*.

**Figure supplement 1.** The construction and genotype of Chi3l1^-/-^ and IEC^ΔChil1^ mice.

**Figure supplement 1—source data 1.** File containing original DNA gels for *Figure 3—figure supplement 1*, indicating the relevant bands.

**Figure supplement 1—source data 2.** Original files for DNA gels for *Figure 3—figure supplement 1*.

**Figure supplement 2.** IEC^ΔChil1^ mice have more abundance of *Turicibacter*.

**Figure supplement 2—source data 1.** Numerical data of *Figure 3—figure supplement 2*.

that mouse Chi3l1 is specifically expressed in intestinal secretory epithelial cells and secreted into the intestinal lumen. Since large amounts of Chi3l1 is secreted into the mucus and Chi3l1 interact with bacteria, we hypothesize that Chi3l1 may regulate the colonization of Gram-positive bacteria in the mucus layer. To test this hypothesis, we isolated bacterial DNA from the ileum and colon mucus of both wildtype and *Chil1^-/-^* mice. Quantification of Gram-positive bacteria and *Lactobacillus* using qPCR revealed that both the ileum and colon mucus of *Chil1^-/-^* mice had significantly lower levels of Gram-positive bacteria and *Lactobacillus* compared to that of wildtype mice (*Figure 4E and F*).

To further validate these results, we labeled a Gram-positive bacteria strain, *Enterococcus faecalis*, with fluorescent D-amino acids (FDAA), which can metabolically label bacterial PGNs (*Wang et al., 2019*). We then performed rectal injection of both wildtype and *Chil1^-/-^* mice with FDAA-labeled *E. faecalis*. The data demonstrated that *Chil1^-/-^* mice had much lower colonization of *E. faecalis* compared to wildtype mice (*Figure 4G*). Besides Gram-positive bacteria, we also performed rectal injection of both wildtype and *Chil1^-/-^* mice with mCherry-OP50 (a strain of *E. coli* that expresses mCherry), we found *Chil1^-/-^* mice had much higher colonization of *E. coli* compared to wildtype mice (*Figure 4—figure supplement 1A*). Based on these findings, we conclude that Chi3l1 promotes the colonization of Gram-positive bacteria, particularly *Lactobacillus*, in the intestinal mucus. Additionally, we also observed that the deletion of Chi3l1 significantly reduced mucus layer thickness, which may be attributed to the disrupted colonization of Gram-positive bacteria in the intestinal mucus layer (*Figure 4—figure supplement 1B and C*).

## Disordered intestinal bacteria in IEC^ΔChil1^ mice contribute to colitis

From the above data, it is evident that Chi3l1 is secreted into the intestinal mucus to influence the colonization of Gram-positive bacteria, particularly *Lactobacillus*. We are now interested in understanding the role of Chi3l1-regulated microbiota in a pathological condition. We observed a significant increase in *Chil1* mRNA expression in the colon tissues of patients with either CD or UC compared to normal tissues (*Figure 5A*). To investigate further, we created a colitis mouse model by subjecting Villin-cre and IEC^ΔChil1^ mice to a 2% dextran sodium sulfate (DSS) diet for 7 days (*Figure 5B*). The severity of colitis was assessed based on weight loss, colon length, and tissue damage. Without the DSS challenge, the colon length and structure were similar between Villin-cre and IEC^ΔChil1^ mice (*Figure 5D and E*). However, upon DSS challenge, the IEC^ΔChil1^ mice showed significantly shorter colon length, faster body weight loss, and more severe inflammation compared to the Villin-cre mice (*Figure 5C–E*).

To rule out the effects of Chi3l1 on the host contributed to colitis, we pretreated the mice with antibiotics to eliminate gut microbiota before inducing colitis (*Figure 5—figure supplement 1A*). The universal bacterial 16S rRNA qPCR data indicated that the majority of gut microbiota were eliminated after antibiotics treatment (*Figure 5—figure supplement 1B*). However, the IEC^ΔChil1^ mice exhibited a milder colitis phenotype, including slower body weight loss, longer colon length, and less inflammation compared to the Villin-cre mice (*Figure 5—figure supplement 1C–E*). We believe that this could

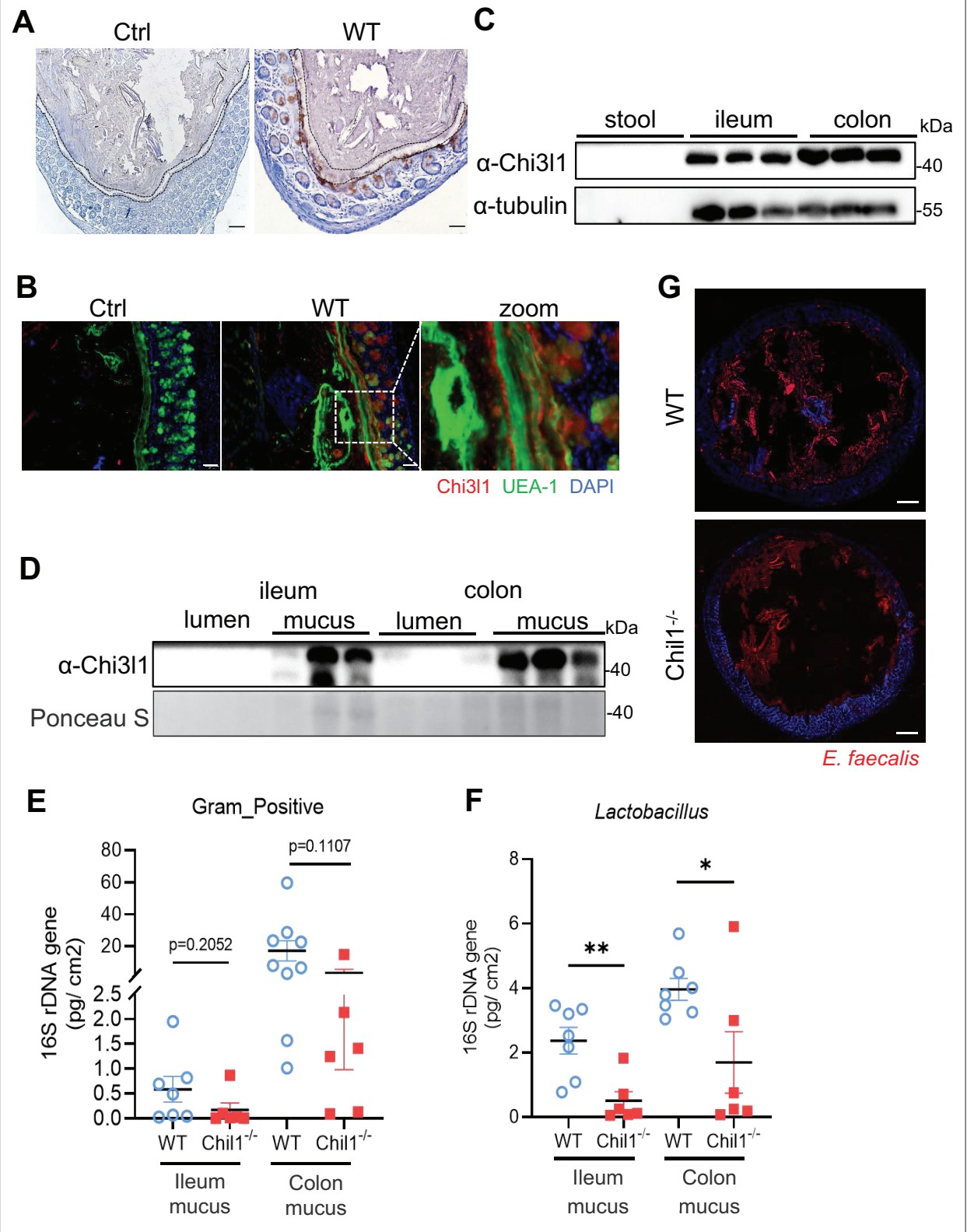

**Figure 4.** Chi3l1 promotes the colonization of Gram-positive bacteria in intestinal mucus layer. (**A**) Immunohistochemical (IHC) staining to detect Chi3l1 in colon mucus layer from wildtype mice. Ctrl (without application of ant-Chi3l1 antibody), WT (wildtype C57BL/6J mice). Black dotted line outlines mucus layer. Scale bars, 50 μm (Ctrl, WT). (**B**) Colons were collected from wildtype mice and stained with UEA-1 (green), Chi3l1 (red), and nuclear DAPI (blue). Ctrl (without application of first antibody), WT (wildtype C57BL/6J mice). Scale bars, 50 μm (Ctrl, WT). (**C**) Stool, ileum, and colon tissues were

*Figure 4 continued on next page*

*Figure 4 continued*

collected from wildtype mice. Western blot was used to detect Chi3l1 expression in these samples. n = 3 mice/sample. (**D**) Both luminal and mucus-associated proteins of either ileum or colon were extracted. Western blot was used to detect Chi3l1 expression in these samples. lumen (luminal proteins), and mucus (mucus-associated proteins). n = 3 mice/sample. (**E, F**) qPCR analysis of specific bacteria in the ileum and colon mucus microbial communities of wildtype and *Chil1⁻/⁻* littermates. (**E**) qPCR analysis of Gram-positive bacteria is shown. (**F**) qPCR analysis of Gram-positive bacteria is shown. Values for each bacterial group are expressed relative to total 16S rRNA levels. WT (wildtype C57BL/6J mice). n = 6–8/group. (**G**) Rectal injection of both wildtype and *Chil1⁻/⁻* mice with FDAA-labeled *E. faecalis* (a Gram-positive bacteria strain) for 4 hr. Colon sections were collected and colonization of *E. faecalis* was examined under microscope. Nuclei were stained with DAPI. Scale bars, 50 μm (WT, *Chil1⁻/⁻*). Representative images are shown in (**A, B, G**), n = 3 mice/group. *p<0.05, **p<0.01, ***p<0.001, ****p<0.0001, ns: no significant difference, error bar indicates SEM.

The online version of this article includes the following source data and figure supplement(s) for figure 4:

**Source data 1.** File containing original western blots for *Figure 4C and D*, indicating the relevant bands.

**Source data 2.** Original files for western blot analysis displayed in *Figure 4C and D*.

**Source data 3.** Numerical data of *Figure 4E and F*.

**Figure supplement 1.** *Chil1⁻/⁻* mice possess shortening mucus layer.

**Figure supplement 1—source data 1.** Numerical data of *Figure 4—figure supplement 1A and B*.

be due to the relationship between Chi3l1 and inflammation. Based on these findings, it is apparent that Chi3l1's effects on gut microbiota play a more significant role in colitis.

To further elucidate the role of Chi3l1-regulated gut microbiota in colitis, we conducted fecal microbiota transplantation (FMT) and *Lactobacillus reuteri* transplantation experiments. We first eliminated gut microbiota through a 10-day course of antibiotics and then performed FMT from Villin-cre mice or administered oral gavage of *L. reuteri* to IEC^ΔChil1 mice for 2 weeks, followed by a 7-day period of 2% DSS feeding (*Figure 5F*). FMT partially restored the colon length of IEC^ΔChil1 mice to that of Villin-cre mice after the DSS challenge, but did not have an impact on body weight loss or the level of inflammation in the gut (*Figure 5G–I*). IEC^ΔChil1 mice transplanted with *Lactobacillus* displayed a similar colitis phenotype as Villin-cre mice, characterized by similar weight loss, colon length reduction, and gut inflammation (*Figure 5G–I*). These findings further validate the notion that Chi3l1-regulated gut microbiota, especially *Lactobacillus*, offers protection against colitis.

## Discussion

The intricate relationship between gut bacteria and their host organisms is crucial for health, maintained through co-evolution and co-speciation. Understanding how the host influences these bacterial communities is critical to understand the complex interplay between host and gut bacteria. In this study, we discovered that Chi3l1 interacts with gut microbiota via PGN, aiding the mucus colonization of beneficial bacteria like *Lactobacillus*, which protects against colitis (*Figure 6*). Our 'scaffold model' demonstrates that Chi3l1 binds to bacterial PGN, helping anchor and organize bacteria within the mucus layer. This interaction promotes the colonization of beneficial bacteria, enhancing gut health.

Several studies have suggested a potential correlation between Chi3l1 and bacteria. For instance, it has been demonstrated that Chi3l1 is produced by IECs in enteritis disease and aids in the elimination of pathogenic bacteria in the gut by neutrophils (*Deutschmann et al., 2019*). Another study observed an increase in Chi3l1 expression in mammary tissues of dairy cows infected with pathogenic *E. coli*, which promoted the recruitment of neutrophils (*Breyne et al., 2018*). Furthermore, Chi3l1 was found to be induced during *Streptococcus pneumoniae* infection, where it enhanced bacterial elimination by preventing the death of lung macrophages and improving host tolerance (*Dela Cruz et al., 2012*). One study demonstrated that Chi3l1 contributes to the pathogenesis of colitis, possibly by facilitating the adhesion and invasion of bacteria onto colonic epithelial cells. However, we observed the presence of the Gram-negative bacterium *Salmonella typhimurium* in the mice used in their study (*Mizoguchi, 2006*). According to standard for SPF-grade mice, *Salmonella* should not be present in laboratory mice, and it was also not detected by our 16S RNA sequencing. However, if mice are inadvertently infected with *Salmonella* in the context of IBD, it could likely exacerbate the development of IBD. Our data also demonstrate that Chi3l1 can bind to Gram-negative bacteria (*Figure 2B*). Therefore, the role of Chi3l1 on microbiota or IBD is heavily dependent on the environmental context. Another study found that caffeine treatment reduce Chi3l1 RNA expression levels, alleviate colitis,

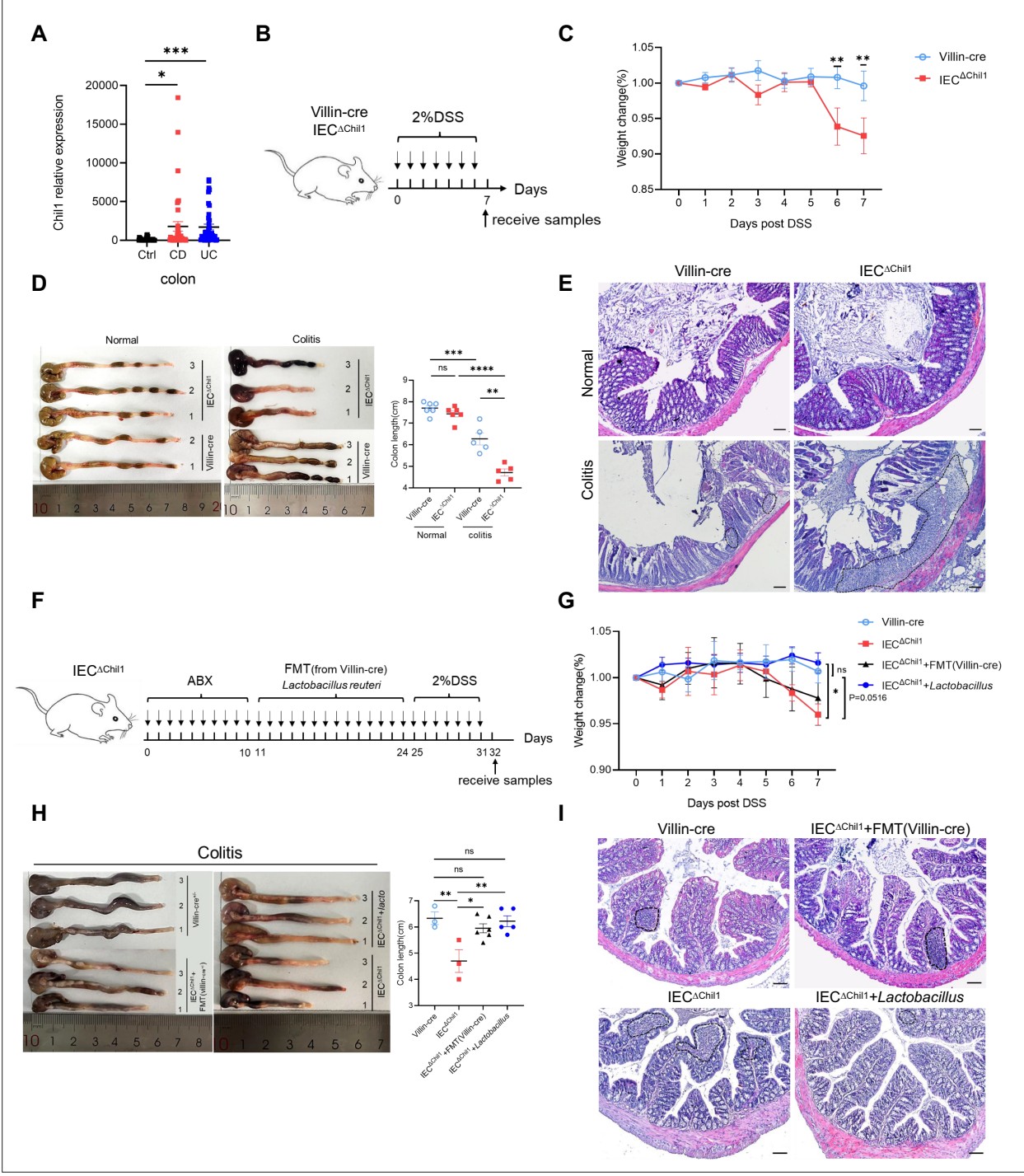

**Figure 5.** Disordered intestinal bacteria in IEC$^{\Delta Chil1}$ mice contribute to inflammatory bowel disease (IBD). (**A**) *Chil1* mRNA relative expression in colon tissues of patients without gut disease (controls, n = 35) or with Crohn's disease (CD, n = 40), ulcerative colitis (UC, n = 40) (GEO datasets: SRP303290). (**B**) Schematic model of the experimental design. Both Villin-cre and IEC$^{\Delta Chil1}$ littermates were fed with 2% dextran sodium sulfate (DSS) in drinking water to induce colitis. (**C**) Weight change of Villin-cre and IEC$^{\Delta Chil1}$ littermates during DSS feeding. Weight change (%) = Current weight/Initial weight. (**D**) Representative colonic length from Normal and DSS-treated Villin-cre and IEC$^{\Delta Chil1}$ littermates (left) and the statistics of colonic length (right). (**E**) H&E staining of mice colon from Normal and DSS-treated Villin-cre and IEC$^{\Delta Chil1}$ littermates. The inflamed areas are outlined by white dotted line, scale bars = 100 μm. (**F**) Schematic of the experimental design. First, antibiotics were used to eliminate gut microbiota for 10 days, and then either fecal microbiota from Villin-cre mice (FMT) or *Lactobacillus reuteri* were transplanted back *to* IEC$^{\Delta Chil1}$ mice orally every day for 2 weeks. Finally, colitis mouse model was constructed by 2% DSS feeding in drinking water for another 7 days. (**G–I**) Villin-cre and IEC$^{\Delta Chil1}$ were only fed with 2% DSS in drinking water for 7 days. IEC$^{\Delta Chil1}$ + FMT(Villin-cre), and IEC$^{\Delta Chil1}$ + *Lactobacillus* were constructed as described in (**F**). (**G**) Weight change of Villin-cre, IEC$^{\Delta Chil1}$, IEC$^{\Delta Chil1}$ +

*Figure 5 continued on next page*

*Figure 5 continued*

FMT(Villin-cre), and IEC$^{\Delta Chil1}$ + *Lactobacillus* mice during DSS feeding. (**H**) Representative colonic length from Villin-cre, IEC$^{\Delta Chil1}$, IEC$^{\Delta Chil1}$ + FMT(Villin-cre), and IEC$^{\Delta Chil1}$ + *Lactobacillus* mice (left) and the statistics of colonic length (right). n = 3–6/group. (**I**) H&E staining of mice colon from Villin-cre, IEC$^{\Delta Chil1}$, IEC$^{\Delta Chil1}$ + FMT(Villin-cre), and IEC$^{\Delta Chil1}$ + *Lactobacillus* mice after DSS treatment. The inflamed area is outlined by black dotted line, scale bars = 100 µm. Representative images are shown in (**C, E, H, I**), n = 3–8 mice/group. *p<0.05, **p<0.01, ***p<0.001, ****p<0.0001, ns: no significant difference, error bar indicates SEM.

The online version of this article includes the following source data and figure supplement(s) for figure 5:

**Source data 1.** Numerical data of *Figure 5A, C, D, G and H* .

**Figure supplement 1.** Chi3l1-mediated bacteria, but not Chi3l1 itself affect more upon the development of colitis.

**Figure supplement 1—source data 1.** Numerical data of *Figure 5—figure supplement 1B*.

and decrease bacterial translocation (***Lee et al., 2014***). However, previous research suggested that caffeine can directly inhibit inflammasome activation by suppressing MAPK/NF-κB signaling pathways (***Vargas-Pozada et al., 2022***). Therefore, caffeine treatment may directly reduce colitis by inhibiting inflammation. Moreover, an additional control group of mice with Chi3l1 specifically knocked out in the gut is needed to examine the role of caffeine in the alleviation of colitis. However, the detailed mechanism of the interaction between Chi3l1 and the gut microbiota remains incompletely understood, partly due to the absence of Chi3l1-specific knockout mice and variations in mouse husbandry conditions. Here, by using intestinal Chi3l1-specific knockout mice, we demonstrated a new function of Chi3l1 in gut, which shapes bacterial colonization through direct interaction with bacterial cell wall component, PGN. Moreover, Chi3l1-regulated gut microbiota, especially *Lactobacillus*, offers protection against colitis.

The bacterial cell wall is a complex structure mainly composed of PGN, but it may also contain other components such as teichoic acids and LTA in Gram-positive bacteria, or an outer membrane containing various polysaccharides, lipids, and proteins in Gram-negative bacteria (***Zhou et al., 2022***). Our findings indicated that LPS, a component of the bacterial cell wall, can slightly increase Chi3l1 expression (***Figure 1F and G***), but higher levels of LPS do not further enhance Chi3l1 expression (data not shown). This suggests that there might be other components of the cell wall that can induce Chi3l1 expression in IECs. Given the structural similarity between chitin and PGN (***Figure 2A***), it is likely that Chi3l1 binds to the polysaccharide chains rather than the tetrapeptide in PGN. Previous studies have investigated the crystal structure of human Chi3l1 in complex with chitin, revealing a binding groove

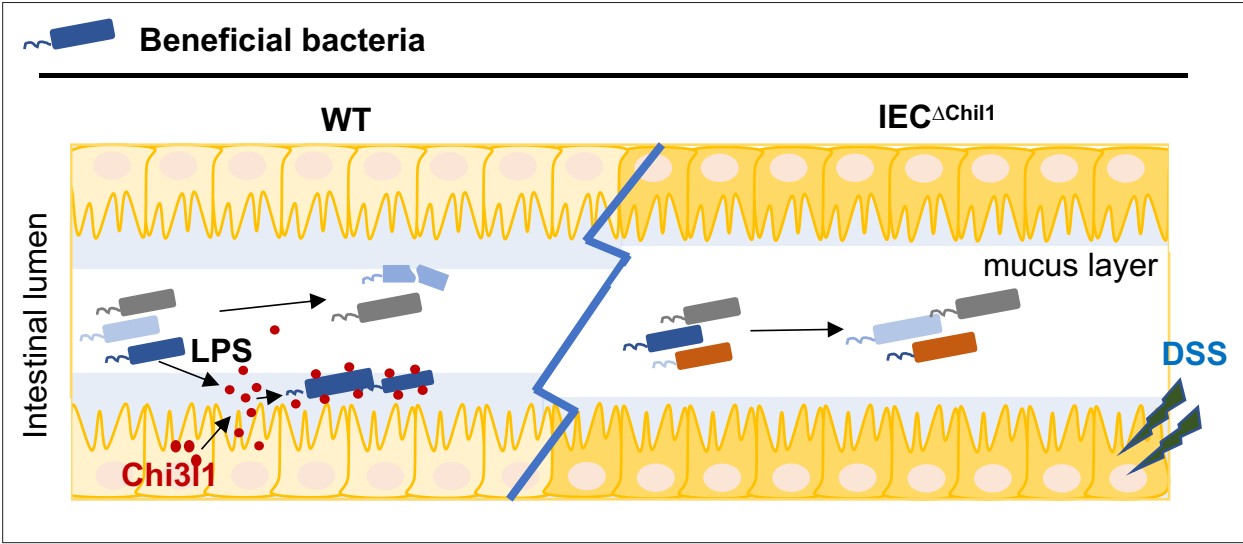

**Figure 6.** A schematic working model. Intestinal epithelial cells are stimulated by the gut microbiota to express Chi3l1. Once expressed, Chi3l1 is secreted into the mucus layer where it interacts with the gut microbiota, specifically through a component of bacterial cell walls called peptidoglycan. This interaction between Chi3l1 and bacteria is beneficial for the colonization of bacteria in the mucus, particularly for Gram-positive bacteria like *Lactobacillus*. Moreover, a deficiency of Chi3l1 leads to an imbalance in the gut microbiota, which exacerbates colitis induced by dextran sodium sulfate (DSS).

with different subsites for chitin fragments (*Fusetti et al., 2003*). Other studies have also identified specific amino acids in chitinases that play a key role in the interaction with chitin (*Ferrandon et al., 2003*; *Ranok et al., 2015*). Nevertheless, further investigation is necessary to better understand the binding sites in Chi3l1 for PGN.

In the colon, there are two distinct parts of the mucus layer. The inner layer is attached and has a low number of microbes, while the outer layer is looser and densely populated by microorganisms. (*Johansson et al., 2011*). The diversity of bacteria in the mucus layer is similar to that found in the gut lumen (*Hu et al., 2021*). It has to be noted that not all Gram-positive bacteria were reduced in *Chil1*[-/-] mice, as we found an increase in *Turicibacter* in the colon (*Figure 3G*) and feces (data not shown). We believe this may be due to a combined effect of the host and gut microbiota. Another important factor for microbial growth in the colon is the integrity of the mucus barrier (*Johansson et al., 2008*). We noticed a thinning of the mucus barrier in the mice lacking Chi3l1 compared to normal mice (*Figure 4—figure supplement 1B and C*). However, we did not observe an increase in mucin-degrading bacteria such as *Bacteroides* or *Allobaculum* (*Glover et al., 2022*; *Raimondi et al., 2021*) or a decrease in mucin-producing cells in these mice. We think that there may be bacteria that aid in the formation of the gut mucus, and these bacteria are decreased in mice lacking Chi3l1. Recent studies have introduced an 'ncapsulation model' regarding the nature of mucus in the colon. According to this model, the mucus in the proximal colon forms a primary encapsulation barrier around fecal material, while the mucus in the distal colon forms a secondary barrier (*Bergstrom et al., 2020*). Our findings indicate that Chi3l1 is expressed throughout the entire colon, including the proximal, middle, and distal sections (data not shown). This suggests that Chi3l1 likely promotes bacterial colonization across the entire colon. Despite most mucus being expelled with feces, the constant production of mucus and the minimal presence of Chi3l1 in feces (*Figure 4C*) indicate that Chi3l1 continuously plays a role in promoting the colonization of microbiota.

In patients with IBD, the density and diversity of the microbial community in the intestines are reduced. Specifically, there is a decrease in *Firmicutes* and an increase in *Bacteroides* and facultative anaerobic bacteria like *Enterobacteriaceae* (*Qin et al., 2010*). However, the cause of colitis in IBD is still a subject of debate. Our data suggest that disruptions in the gut microbiota contribute to colitis. In summary, our study demonstrates that bacterial challenge induces the expression of Chi3l1 in IECs. Once produced, Chi3l1 is released into the mucus layer where it interacts with the gut microbiota, particularly through PGN, a primary component of bacterial cell walls. This interaction is beneficial for the colonization of bacteria, especially Gram-positive bacteria like *Lactobacillus*, in the mucus layer. Dysbiosis resulting from a lack of Chi3l1 exacerbates DSS-induced colitis, highlighting the role of dysbiosis as a contributing factor to colitis.

# Materials and methods

### Key resources table

| Reagent type (species) or resource | Designation | Source or reference | Identifiers | Additional information |
|---|---|---|---|---|
| Cell line (*Homo sapiens*) | DLD-1 cells | ATCC | CCL-221 | |
| Recombinant DNA reagent | PLKO.1-Puro (plasmid) | Addgene | RRID:Addgene_10878 | Pol III-based shRNA backbone |
| Transfected construct (human) | sh*Chil1* (constructed from pLKO.1 – TRC) | This paper | Constructed from RRID:Addgene_10878 | Lentiviral construct to transfect and express the shRNA |
| Strain, strain background (*Staphylococcus saprophyticus*) | *Staphylococcus saprophyticus* | ATCC | 15305 | |
| Strain, strain background (*Enterococcus faecalis*) | *E. faecalis* | ATCC | 33186 | |
| Strain, strain background (*Lactobacillus reuteri*) | *Lactobacillus reuteri* | ATCC | 23272 | |

*Continued on next page*

*Continued*

| Reagent type (species) or resource | Designation | Source or reference | Identifiers | Additional information |
|---|---|---|---|---|
| Strain, strain background (*Escherichia coli*) | K12 | Dharmacon | Cat# OEC5042 | |
| Strain, strain background (*E. coli*) | OP50 | CGC | RRID:WB-STRAIN:WBStrain00041969 | |
| Strain, strain background (*E. coli*) | OP50-mCherry | Provided by Bin Qi Lab; *He et al., 2023* | | |
| Strain, strain background (*Staphylococcus sciuri*) | *Staphylococcus sciuri* | Identified from C57BL/6J wildtype mice stools | This paper | This strain is used in *Figure 1D* |
| Strain, strain background (*E. coli*) | E. coli | Identified from C57BL/6J wildtype mice stools (this paper) | This paper | This strain is used in *Figure 1D* |
| Antibody | Anti-Chi3l1 (rabbit polyclonal) | Invitrogen | PA5-95897 RRID:AB_2807699 | IHC (1:200) |
| Antibody | Anti-Chi3l1 (rabbit polyclonal) | Abcam | ab180569 RRID:AB_2891040 | IF (1:400) |
| Antibody | Anti-UEA-1-FITC (*Ulex europaeus*) | GeneTeX | GTX01512 | IF (1:200) |
| Antibody | Anti-Cha-A (mouse monoclonal) | Santa Cruz | sc-393941 RRID:AB_2801371 | IF (1:200) |
| Antibody | Anti-mouse Chi3l1 Purified Rat monoclonal IgG (rat monoclonal) | R&D | MAB2649 RRID:AB_2081263 | WB (1:2000) |
| Antibody | Anti-MUC2 (rabbit polyclonal) | Invitrogen | PA5-103083 RRID:AB_2852453 | IF (1:50) |
| Antibody | Gram-positive bacteria LTA (mouse monoclonal) | Invitrogen | MA1-7402 RRID:AB_1017302 | IF (1:50) |
| Antibody | 488-conjugated Affinipure Goat Anti-Rabbit (goat polyclonal) | Jackson | 111-545-003 RRID:AB_2338046 | IF (1:1000) |
| Antibody | Immunoresearch AlexaFluor 594 AffiniPure Goat Anti-Mouse IgG (H+L) (goat polyclonal) | Jackson | 115-585-003 RRID:AB_2338046 | IF (1:1000) |
| Antibody | Goat anti-Rabbit IgG (H+L) Secondary Antibody-Biotin (goat polyclonal) | Invitrogen | 65-6140 RRID:AB_2533969 | IHC (1:2000) |
| Antibody | Horseradish peroxidase conjugate antibody | Invitrogen | A2664 RRID:AB_2764530 | IHC (1:2000) |
| Antibody | Rabbit anti-Chi3l1 antibody | Proteintech | 12036-1-AP RRID:AB_2877819 | IF (1:200) WB (1:2000) |
| Antibody | Goat anti-Rabbit IgG (goat polyclonal) | Jackson ImmunoResearch | 111-035-0030 | WB (1:10,000) |
| Antibody | Anti-alpha-Actinin (mouse IgG1) | Cell Signaling | 69758S | WB (1:1000) |
| Antibody | Anti-Rat-IgG (goat) | Cell Signaling | 7077S | WB (1:10,000) |
| Antibody | Goat anti-mouse | Invitrogen | 62-6520 RRID:AB_2533947 | WB (1:10,000) |
| Antibody | Mouse anti-α-tubulin antibody (mouse monoclonal) | Sigma | T5168 RRID:AB_477579 | WB (1:2000) |
| Chemical compound, drug | Dextran sodium sulfate salt | MP | CAS:9011-18-1 | |
| Chemical compound, drug | Hematoxylin | Servicebio | G1004 | |
| Chemical compound, drug | Eosin | Biosharp | BL703b | |
| Chemical compound, drug | Tris | Solarbio | 77-86-1 | |
| Chemical compound, drug | Sodium chloride | Solarbio | 7647-14-5 | |
| Chemical compound, drug | Disodium salt dihydrate (EDTA) | Sangon Biotech | 6381-92-6 | |

*Continued on next page*

*Continued*

| Reagent type (species) or resource | Designation | Source or reference | Identifiers | Additional information |
|---|---|---|---|---|
| Chemical compound, drug | Sodium dodecyl sulfate (SDS) | BBI | A601336-0500 | |
| Chemical compound, drug | Egtazic acid, glycol ether diamine tetraacetic acid (EGTA) | BBI | 67-42-5 | |
| Chemical compound, drug | TritonX-100 | BBI | 9002-93-1 | |
| Chemical compound, drug | Citric acid | Sangon Biotech | 77-92-9 | |
| Chemical compound, drug | Vancomycin | Solarbio | V8050 | |
| Chemical compound, drug | Ampicillin-sodium salt | Solarbio | A8180 | |
| Chemical compound, drug | Metronidazole | Solarbio | M8060 | |
| Chemical compound, drug | Neomyein sulfate | Solarbio | N8090 | |
| Chemical compound, drug | MRS broth | Solarbio | M8540 | |
| Chemical compound, drug | Goat serum | Solarbio | | |
| Commercial assay or kit | AB-PAS staining kit | Solarbio | G1285 | |
| Commercial assay or kit | EIANamp stool DNA kit | TIANGEN | DP328-02 | |
| Commercial assay or kit | PAGE Gel Fast Preparation Kit | Shanghai Epizyme biotechnology | PG113 | |
| Commercial assay or kit | Gel extraction kit | Omega | D2500-02 | |
| Commercial assay or kit | DAB Substrate Kit | ZSGB-BIO | zli-9018 | |
| Other | SYBR Green kit | Thermo Fisher | A25742 | Using for qPCR |
| Other | FDAA | 5TAMRA; CHINESE PEPTIDE | CS-11-00433 | Using for label *E. faecalis* |
| Other | Antifade mounting medium | Vectashield | H-1000-10 | Using for IF staining |

## Animal experiments and procedures

C57BL/6J (strain no. N000013), germ-free (strain no. N000295), *Chil1*$^{fl/fl}$ (strain no. T013652), and *Chil1*$^{-/-}$ (strain no. T014402) mice were purchased from GemPharmatech. Villin-cre mice were provided by Dr. Qun Lu (Yunnan University, China). All mouse colonies were maintained at the animal core facility of Yunnan University. C57BL/6J was used as wildtype control since *Chil1*$^{-/-}$ mice are on the C57BL/6J background, as determined by PCR (data not shown). The animal studies described have been approved by the Yunnan University Institutional Animal Care and Use Committee (IACUC, approval no. YNU20220256). Female mice aged 8–10 weeks old were used in most studies.

### Genotyping

Tail clippings were placed in 1.5 mL tubes with 75 µL of master mix solution (60 µL $H_2O$, 7.5 µL 250 mM NaOH, 7.5 µL 2 mM EDTA) and incubated at 98°C for 1 hr. After cooling to 15°C, 75 µL of neutralization buffer (40 mM Tris–HCl, pH 5.5) was added, and the samples were centrifuged at 4000 rpm for 3 min. A 1:10 dilution was prepared by mixing 2 µL of supernatant with 18 µL of water for genotyping PCR. A 25 µL PCR reaction mix was prepared with 12.5 µL of 2× Taq Master Mix (Dye Plus, Vazyme P112-03), 1 µL of each primer, 2 µL of template, and water. PCR was conducted using a Bio-Rad machine with the following program: 95°C for 5 min; 98°C for 30 s; 65°C with a 0.5°C decrement per cycle for 30 s; 72°C for 45 s, repeating steps 2–4 for 20 cycles; 98°C for 30 s; 55°C for 30 s; 72°C for 45 s, repeating steps 5–7 for 20 cycles; 72°C for 5 min; and hold at 10°C. Primer sequences are listed in ***Supplementary file 1***.

### Rectal administration of FDAA-labeled *E. faecalis*

For FDAA-labeled *E. faecalis*, *E. faecalis* were grown in LB media until reaching mid-exponential phase, approximately 4 hr. FDAA was then added to the culture media to a final concentration of about 17 μM. The *E. faecalis* continued to grow for 4 hr and were then harvested by centrifugation (5000 × *g* for 10 min), washed, and resuspended in PBS at a density of 5 × 10⁹ CFU/mL. For rectal administration, wildtype and *Chil1⁻/⁻* mice were fasted overnight (5:00 pm to 9:00 am) before intraperitoneally (i.p.) injecting them with 400 mg/kg tribromoethanol (Nanjing AIBI BioTechnology, M2910), followed by rectal injection of 1 × 10⁹ FDAA-labeled *E. faecalis* in 200 μL PBS via a flexible catheter. The catheter was inserted into the anus to a depth of 4.5 cm, and the *E. faecalis* were injected slowly to avoid overflow. The mice were then kept upside down for approximately 2 min. After 4 hr of rectal injection, the mice's colons were collected and immediately embedded in OTC embedding medium for observation on 5-μm-thick frozen sections. OP50-mCherry (1 × 10⁹ CFU/mice) were treated same as *E. faecalis*.

### DSS-induced mouse colitis

Villin-cre and IEC$^{ΔChil1}$ mice were fed a 2% DSS (MP, 9011-18-1) solution in their drinking water for 7 days. The mice's body weight was monitored daily during the feeding period. After the 7-day DSS treatment, the mice were sacrificed. Colons were collected, and their length was measured from the cecum to the rectum. Colon paraffin sections were harvested, and H&E staining was performed to examine gut inflammation.

### Antibiotics treatment

IEC$^{ΔChil1}$ mice were fed an antibiotics mixture containing 0.5 mg/mL of metronidazole (Solarbio, M8060), 1 mg/mL of vancomycin (Solarbio, V8050), 1 mg/mL of ampicillin (Solarbio, A8180), and 0.5 mg/mL of neomycin sulfate (Solarbio, N8090) in their drinking water for 7 days (approximately 5 mL per mouse per day). After the 7-day antibiotics treatment, the mice were orally gavaged with 200 μL of the antibiotics mixture for another 3 days. Microbiota depletion was examined in feces using 16S rRNA qPCR on the tenth day of antibiotics feeding.

### Fecal microbiota transplantation (FMT)

Fresh feces were collected from 8- to 10-week-old Villin-cre mice and immediately snap-frozen in liquid nitrogen. On the experimental day, feces were resuspended in PBS to a concentration of 200 mg/mL and centrifuged at 350 × *g* for 5 min to collect the supernatant. Antibiotics pre-treated IEC$^{ΔChil1}$ mice were orally gavaged with 10 μL of the dissolved feces per gram of mouse weight for 14 days.

### Oral gavage of *Lactobacillus reuteri*

*L. reuteri* was grown in MRS broth at 37°C for 48 hr under anaerobic conditions. The bacteria were harvested by centrifugation (5000 × *g* for 10 min), washed, and resuspended in PBS at a density of $OD_{605}$ = 1.2–1.3/mL. Antibiotics pre-treated IEC$^{ΔChil1\ mice}$ were orally gavaged with 200 μL of the dissolved *L. reuteri* per mouse for 14 days.

### **Bacterial speciesidentification**

Fresh feces were collected from 8-week-old wildtype mice and dissolved in LB culture medium. The dissolved feces were then cultured at 37°C for 12 hr and plated onto LB agar plates. The grown colonies were picked using sterile pipette tips and resuspended in 20 μL of sterile water. A PCR reaction was performed using 2 μL of the bacterial suspension as template DNA and universal bacterial 16S rRNA primers (27F, 5'-AGAGTTTGATCCTGGCTCAG-3' and 1492R, 5'-GGTTACCTTGTTACGACTT-3') with reaction conditions: 95°C for 5 min followed by 35 cycles of 95°C for 30 s, 55°C for 30 s, 72°C for 2 min and then 72°C for 20 min. The amplicons were then sequenced, and the resulting sequences were analyzed using BLASTN and the NCBI database for species identification.

### **Treatment of DLD-1 cells with live, heat-killed bacteria or LPS**

Bacteria harvested from mouse feces or specific strains were grown in LB medium at 37°C for 12 hr under aerobic conditions. The bacteria were then collected by centrifugation (5000 × *g* for 10 min),

washed, and resuspended in PBS at a density of $1.2 \times 10^{10}$ CFU/mL. Live bacteria were used directly, while heat-killed bacteria required further heating at 80°C for 30 min. DLD-1 cells were grown in DMEM supplemented with 10% FBS. Prior to treatment, the cells were replated and allowed to reach 80% confluency. Either live or heat-killed bacteria were added to the cells at a multiplicity of infection (MOI) of 20 each well, and 100 pg/mL LPS treatment (Sigma, L4392; diluted in PBS) was also performed. After 12 hr incubation, cellular proteins were extracted for western blot analysis or the cells were subjected to immunofluorescent staining.

## Cell culture and bacterial adhesion assay

DLD-1 cells (ATCC CCL-221) were cultured in six-well plates until they reached approximately 80% confluency per well. The cells were then washed three times with PBS to remove any residual antibiotics. Fresh DMEM supplemented with 10% FBS (without antibiotics) was added to the cells. Absence of mycoplasma contamination was confirmed using PCR. *E. coli* strains were introduced to each well at an MOI of 20 and incubated for 2.5 hr at 37°C. After incubation, the cells were washed three times with PBS. For the adhesion assays, the cells were lysed with 1 mL of 1% Triton-X100 (BBI, 9002-93-1) in deionized water for 30 min. The cell lysates were then plated onto Luria broth (LB) agar plates at various dilutions and incubated overnight at 37°C. The following day, colonies were counted and calculated as CFU/mL.

## shRNA lentivirus packaging and transfection

For the packaging of shRNA lentivirus, shRNA control and Chi3l1 shRNA1 lentiviruses were packaged in 293T cells. The cell culture medium containing the virus was collected and stored at –80°C. For transfection of DLD-1 cells, the cells were cultured in six-well plates until they reached approximately 80% confluency. The cell culture medium was then replaced with a mixture of culture medium containing the virus and fresh cell culture medium in a 1:1 ratio. Following transfection, the bacterial adhesion assay was performed as described above.

## Bacterial or peptidoglycan binding assay

Different bacteria strains were grown in LB medium at 37°C for 12 hr under aerobic conditions. The bacteria were collected by centrifugation ($5000 \times g$ for 10 min), washed, and resuspended in MES buffer (25 mM MES, 25 mM NaCl, pH = 6.0) at a density of $5 \times 10^9$ CFU/mL. 1 µg recombinant mouse Chi3l1 (rmChi3l1) was added to the bacterial suspension and incubated at 4°C under rotation overnight. Supernatant, wash fractions, and bacterial-bound fractions were collected and analyzed using western blot analysis. For the PGN binding assay, 1 µg rmChi3l1 or recombinant human Chi3l1 (rhChi3l1) and BSA was incubated with 100 µg PGN. The incubation and wash procedure were similar to the bacterial binding assay, and the proteins in each fraction were analyzed using silver staining or western blot.

## Hematoxylin/eosin (H&E) and Periodic acid–Schiff and Alcian blue (AB-PAS) staining

Tissues were fixed with buffered 10% paraformaldehyde (BI, E672001-0500) overnight at 4°C and embedded in paraffin. Ultra-thin tissue slices (5 µm) were prepared and deparaffinized. H&E staining was performed on the tissue sections, and slides were examined under a microscope (Leica). For AB-PAS staining, tissues were fixed in Carnoy's solution (60% ethanol, 30% chloroform, 10% acetic acid) for 24 hr at 4°C and embedded in paraffin. AB-PAS staining (Solarbio, G1285) was performed according to the manufacturer's protocol. The staining was visualized under the microscope (Leica).

## Immunohistochemical (IHC) and immunofluorescent (IF) staining

Tissue paraffin sections were prepared as previously described for H&E staining. Antigen retrieval was performed by treating the sections with citric acid (pH 6.0) at 95°C for 15 min, followed by cooling to room temperature. The sections were then washed with PBS and ddH$_2$O. To block any nonspecific binding, a blocking buffer containing 5% goat serum, 3% BSA, and 0.1% Triton X-100 in PBS was applied to the sections for 1 hr at room temperature in a humidity chamber. The sections were then incubated with anti-Chi3l1 primary antibodies (Invitrogen, PA5-95897, 1:200), followed by staining with Goat anti-Rabbit IgG (H+L) Secondary Antibody-Biotin (Invitrogen, 65-6140, 1:2000). Finally, the

sections were stained with Horseradish Peroxidase conjugate antibody (Invitrogen, A2664, 1:2000) and developed with DAB for 10 min. The slides were examined under a microscope (Leica).

Immunofluorescent staining was also performed on paraffin sections using specific primary antibodies. These included anti-MUC2 (Invitrogen, PA5-103083, 1:50), anti-Chi3l1 (Abcam, ab180569, 1:400; antigen retrieval with Tris-EDTA at pH 9.0), anti-ChgA (Santa Cruz, sc-393941, 1:200; antigen retrieval with citrate at pH 6.0), anti-UEA-1-FITC (GeneTex, GTX01512, 1:200; antigen retrieval with citrate at pH 6.0), and anti-LTA (Invitrogen, MA1-7402, 1:50). The secondary antibodies used were 488-conjugated Affinipure Goat Anti-Rabbit IgG(H+L) (Jackson ImmunoResearch, 111-545-003, 1:1000) and 594-conjugated Goat Anti-Mouse IgG (H+L) (Jackson ImmunoResearch, 115-585-003, 1:1000). Slides were washed and mounted with antifade medium (Vectashield, H-1000-10). Nuclei were stained with DAPI (Beyotime, c1006, prediluted). Images were captured using a fluorescence microscope (Leica, 2084 DP-80).

For immunofluorescent staining on DLD-1 cells, cells were seeded on coverslips in 12-well plate and challenged with 100 pg/mL LPS (Sigma, L4391; diluted in PBS) for 12 hr. Cells were washed with cold PBS twice gently, then fixed with 2% paraformaldehyde in PBS at room temperature for 10 min. After removal of fixation buffer and wash twice with cold PBS, cells were blocked with blocking buffer (3% BSA, 0.5% Triton-X-100 in PBS) for 1 hr in humidity chamber at room temperature. Rabbit anti-Chi3l1 antibody (Proteintech, 12036-1-AP; 1:200) was applied at room temperature for 1 hr. After wash with 1× TBST three times, secondary antibody AlexaFluor 488 (Jackson ImmunoResearch, 111-545-003, 1:1000) was applied for another 1 hr at room temperature in a humidified chamber under darkness. Finally, cells were counterstained with DAPI (Beyotime, C1006) and mounted onto slides. Images were captured using an Olympus BX53F2 microscope.

## Fluorescence in situ hybridization (FISH)

Murine intestinal paraffin sections were prepared according to the previously described method for H&E staining. The tissues sections were rehydrated using a graded ethanol series and then washed with distilled water. The Gram-positive bacterial probe, consisting of three different sequences (/5Alex550N/TGGAAGATTCCCTACTGC/3AlexF550N/, /5Alex550N/CGGAAGATTCCCTACT GC/3AlexF550N/, /5Alex550N/CCGAAGATTCCCTACTGC/3AlexF550N/), or the control nonspecific probe (/5Alex550N/ACTCCTACGGGAGGCAGC/3AlexF550N/), was diluted to a concentration of 100 nM in FISH hybridization buffer (containing 20 mM Tris pH 7.2, 0.9 M NaCl, and 0.1% SDS) and applied to the slides. The slides were then incubated overnight at 56°C in a humidified chamber. Following incubation, the slides were washed and the nuclei were counterstained with DAPI. The images were captured using a fluorescence microscope (Leica, 2084 DP-80).

## Immunoblot and silver staining

Protein extraction from cultured cells involved lysing the cells in 2% SDS lysis buffer, which is prepared by dissolving 2 g of SDS powder in 100 mL of sterilized ddH$_2$O. Bacteria or PGN precipitates were resuspended in MES buffer, which contained 25 mM MES, 25 mM NaCl, and had a pH of 6.0. For protein extraction from mice ileum and colon tissues, 30 mg of snap-frozen tissues were homogenized in 1 mL of RIPA buffer, which contained 10 mM Tris–HCl (pH 8.0), 1 mM EDTA, 0.5 mM EGTA, 1% TritonX-100, 0.1% sodium deoxycholate, 0.1% SDS, 140 mM NaCl, 1 mM PMSF, and a proteinase inhibitor. The lysates were then supplemented with 5× SDS loading buffer to a final concentration of 1×. The resulting mixture was subsequently boiled at 100°C for 10 min and centrifuged at 4°C and 12,000 rpm for 10 min. The supernatants were collected for western blot analysis. The supernatants were separated using a 10% SDS-PAGE gel and then transferred to a polyvinylidene fluoride membrane. The membranes were blocked with 5% nonfat milk in TBST buffer (containing 0.1% Tween-20 in Tris-buffered saline) and sequentially incubated with primary antibodies and appropriate horseradish peroxidase (HRP)-conjugated secondary antibodies. Protein bands were detected using enhanced chemiluminescence (ECL) reagent with a Minichemi Chemiluminescence Imaging System. The primary antibodies used included anti-Chi3l1 (RD, MAB2649, 1:2000), anti-alpha-Actinin (Cell Signaling, 69758S, 1:1000), rabbit anti-Chi3l1 antibody (Proteintech, 12036-1-AP, 1:2000), and mouse anti-α-tubulin antibody (Sigma, T5168, 1:2000). The secondary antibodies used were goat anti-Rat-IgG (Cell Signaling, 7077S, 1:10,000), goat anti-mouse (Invitrogen, 62-6520, 1:10,000) and Goat anti-Rabbit IgG (Jackson ImmunoResearch, 111-035-0030, 1:10,000). For silver staining, the PAGE

gel was subjected to silver staining using a fast-silver staining kit (Beyotime, P0017S) following the manufacturer's instructions.

## DNA extraction for 16S rRNA analysis

For isolation of luminal contents from murine ileum and colon, a 9 cm section of ileum and 3 cm section of colon were cut open longitudinally, and luminal contents were scratched off into a pre-weighed 2 mL sterile freezing vial. The weight of the contents was recorded for further processing. For isolation of mucus from murine ileum and colon, tissues were flushed with 2 mL of ice-cold PBS into a pre-weighed 2 mL sterile freezing vial after removal of the liminal contents. The mucus was then pelleted by centrifugation at $10,000 \times g$ for 10 min, and the supernatant was removed. For feces isolation, fresh murine feces were collected into a sterile freezing vial weighing 2 mL and immediately snap-frozen in liquid nitrogen. For bacterial DNA extraction, DNA was extracted and purified following the manufacture's protocol using the ElANamp stool DNA kit (TIANGEN, DP328-02).

## Microbiota 16S rRNA gene sequencing

Fecal samples, ileum contents, and colon contents were collected from wildtype, $Chil1^{-/-}$ littermates, or Villin-cre, IEC$^{\Delta Chil1}$ littermates and immediately frozen in liquid nitrogen. The microbial genomic DNA was extracted and 16S rRNA sequencing was performed by Biomarker Technologies. The hypervariable regions V3 and V4 of the bacterial 16S rRNA gene were sequenced using universal primers that flank these regions V3 (338F 5'-ACTCCTACGGGAGGCAGCA-3') and V4 (806R 5'- GGAC TACHVGGGTWTCTAAT-3'). The sequencing was done using the IIlumina Sequencing platform. The resulting 16S rRNA gene sequences were analyzed using scripts from the BMK Cloud platform (https://www.biocloud.net/). The microbial classification was performed using the SILVA138 and hierarchical clustering algorithms. The OTUs were determined by clustering the sequences with 97% similarity and were classified into different taxonomic ranks. The relative abundance of each bacterial species was visualized using R software. The raw 16S rRNA gene sequencing data can be accessed on the BMKCloud platform under the project ID Microbial_updateReport_20211221092022313.

## 16S qPCR analysis

Quantitative PCR was performed using SYBR green master mix (Thermo Fisher, A25742) in triplicates. This was done on a Real-Time PCR QuatStudio1 with accompanying software, following the instructions provided by the manufacturer (Life Technologies, Grand Island, NY). The abundance of specific bacterial groups in the intestine was determined using qPCR with either universal or bacteria-specific 16S rRNA gene primers. Standard curves were constructed with *E. coli OP50* 16S rRNA gene, which was amplified using conserved 16S rRNA primers. It should be noted that qPCR measures the number of 16S gene copies per sample, not the actual bacterial numbers or colony forming units. Primer sequences are provided in *Supplementary file 1*.

## Statistical analysis

Data were presented as mean ± SEM. Statistical analyses were carried out using GraphPad Prism (GraphPad Software). Comparisons between two groups were carried out using unpaired Student's *t*-test. Comparisons among multiple groups (n ≥ 3) were carried out using one-way or two-way ANOVA. Data are presented as mean ± SEM, and $p<0.05$ was considered a significant difference, *$p<0.05$, **$p<0.01$, ***$p<0.001$, ****$p<0.0001$, 'ns' represents no significant difference.

## Acknowledgements

We thank Qun Lu (Yunnan University) for providing Villin-cre mice. We thank Wenxiang Fu (Yunnan University) for imaging technical support. We thank Jianwei Sun (Yunnan University) for providing DLD-1 cell line. We thank Zehan Hu (Shanghai Jiao Tong University) for methods of bacteria 16S FISH staining. This work was supported by the Ministry of Science and Technology of China (2019YFA0803100, 2019YFA0802100 to BQ), National Natural Science Foundation of China (32071129 to ZS, 32170794 to BQ), Yunnan Fundamental Research Projects (202101AT070022, 202001AW070006 to ZS, 202201AT070196 to BQ), Science and Technological Talent Cultivation Plan of Yunnan Province (C619300A086 to ZS, K264202230211 to BQ), and Yunnan Provincial Science and Technology Project at Southwest United Graduate School (202302AP370005 to BQ).

# Additional information

## Funding

| Funder | Grant reference number | Author |
| --- | --- | --- |
| Ministry of Science and Technology of the People's Republic of China | 2019YFA0803100 | Bin Qi |
| National Natural Science Foundation of China | 32071129 | Zhao Shan |
| National Natural Science Foundation of China | 32170794 | Bin Qi |
| Yunnan Provincial Science and Technology Department | 202101AT070022 | Zhao Shan |
| Yunnan Provincial Science and Technology Department | 202201AT070196 | Bin Qi |
| Science and Technological Talent Cultivation Plan of Yunnan Province | C619300A086 | Zhao Shan |
| Science and Technological Talent Cultivation Plan of Yunnan Province | K264202230211 | Bin Qi |
| Yunnan Provincial Science and Technology Project at Southwest United Graduate School | 202302AP370005 | Bin Qi |
| Ministry of Science and Technology of the People's Republic of China | 2019YFA0802100 | Bin Qi |
| Yunnan Provincial Science and Technology Department | 202001AW070006 | Zhao Shan |

The funders had no role in study design, data collection and interpretation, or the decision to submit the work for publication.

## Author contributions

Yan Chen, Resources, Data curation, Formal analysis, Investigation, Visualization, Methodology, Writing – original draft, Project administration; Ruizhi Yang, Resources, Data curation, Formal analysis, Validation, Investigation, Visualization, Methodology, Writing – original draft; Bin Qi, Conceptualization, Supervision, Funding acquisition, Project administration, Writing - review and editing; Zhao Shan, Conceptualization, Resources, Supervision, Funding acquisition, Validation, Investigation, Visualization, Methodology, Writing – original draft, Project administration, Writing - review and editing

## Author ORCIDs

Bin Qi ⓘ https://orcid.org/0000-0003-2261-1550
Zhao Shan ⓘ https://orcid.org/0000-0001-5064-1023

## Ethics

The animal studies described have been approved by the Yunnan University Institutional Animal Care and Use Committee (IACUC, Approval No. YNU20220256).

Reviewer #1 (Public review): https://doi.org/10.7554/eLife.92994.3.sa1
Reviewer #2 (Public review): https://doi.org/10.7554/eLife.92994.3.sa2
Author response https://doi.org/10.7554/eLife.92994.3.sa3

## Additional files

### Supplementary files
- MDAR checklist
- Supplementary file 1. Genotyping and qPCR Primers.

### Data availability

Sequencing data have been deposited in NCBI under accession codes PRJNA1155018 and PRJNA1157557. All data generated or analyzed during this study are included in the manuscript and supporting files; source data files have been provided for all figures.

The following datasets were generated:

| Author(s) | Year | Dataset title | Dataset URL | Database and Identifier |
|---|---|---|---|---|
| Yan C | 2024 | Identification of intestinal contents and fecal bacteria in mice | https://www.ncbi.nlm.nih.gov/bioproject/PRJNA1155018/ | NCBI BioProject, PRJNA1155018 |
| Rui Z | 2024 | Bacterial microbial diversity in mouse stool | https://www.ncbi.nlm.nih.gov/bioproject/PRJNA1157557/ | NCBI BioProject, PRJNA1157557 |

The following previously published dataset was used:

| Author(s) | Year | Dataset title | Dataset URL | Database and Identifier |
|---|---|---|---|---|
| D'alessio S, Ungaro F, Massimino L, Lamparelli LA, Danese S | 2021 | Characterization of the IL23-IL17 immune axis in IBD patients | https://www.ncbi.nlm.nih.gov/geo/query/acc.cgi?acc=GSE165512 | NCBI Gene Expression Omnibus, GSE165512 |

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
