## [Editor Report · eLife assessment]

Supported by **convincing** data, this **valuable** study demonstrates that the Chitinase 3-like protein 1 (Chi3l1) interacts with gut microbiota and protects animals from intestinal injury in laboratory colitis model. The revised article sufficiently addressed the reviewers' comments. The work will be of interest to scientists studying crosstalk between gut microbiota and inflammatory diseases.

---

## [Referee Report · Reviewer #1 (Public review)]

The manuscript by Chen et al. investigated the interaction between CHI3L1, a chitinase-like protein in the 18 glycosyl hydrolase family, and gut bacteria in the mucosal layers. The authors provided evidence to document the direct interaction between CHI3L1 and peptidoglycan, a major component of bacterial cell wall. Doing so, Chi3l1 produced by gut epithelial cells regulates the balance of gut microbiome and diminishes DSS-induced colitis, potentially through the colonization of protective gram-positive bacteria such as lactobacillus.

The study is the first to systemically document the interactions between Chi3L1 and microbiome. Convincing data were shown to characterize the imbalance of gram-positive bacteria in the newly generated gut epithelial-specific Chi3L1 deficient mice. Comprehensive FMT experiments were performed to demonstrate the contributions of gut microbiome using the mouse colitis model. The manuscript is strengthened by additional mechanistic studies concerning the binding between Chi3l1 and peptidoglycan, and discussions on existing body of literature demonstrating that detrimental roles of Chi3l1 in mouse IBD model, which conflict with the current study.

---

## [Referee Report · Reviewer #2 (Public review)]

Chen et al. investigated the regulatory mechanism of bacterial colonization in the intestinal mucus layer in mice and its implications to intestinal diseases. They demonstrated that Chi3l1 is a protein produced and secreted by intestinal epithelial cells into the mucus layer upon response to the gut microbiota, which has a turnover effect on facilitating the colonization of gram-positive bacteria in the mucosa. The data also indicate that Chi3l1 interacts with the peptidoglycan of the bacteria cell wall, supporting the colonization of beneficial bacteria strains such as Lactobacillus, and that deficiency in Chi3l1 predisposes mice to colitis. The inclusion of a small but pertinent piece of human data added to solidify their findings in mice.

Overall, the experiments were appropriately designed and executed with precision. The revised manuscript represents a significant improvement over the initial version. The inclusion of new, higher-resolution images provides stronger support for the conclusions drawn. Additionally, statistical analyses of the imaging data, as recommended, have been integrated. The authors have effectively addressed the majority of the reviewers' suggestions and criticisms, making this version well-suited for publication.

---

## [Author Response]

The following is the authors’ response to the original reviews.

**Public Reviews:**
**Reviewer 1**:(1) In Figure 1, it is curious that the authors only chose *E.coli* and staphytlococcus sciuri to test the induction of Chi3l1. What about other bacteria? Why does only *E. coli* but not staphytlococcus sciuri induce chi3l1 production? It does not prove that the gut microbiome induces the expression of Chi3l1. If it is the effect of LPS, does it trigger a cell death response or inflammatory responses that are known to induce chi3l1 production? What is the role of peptidoglycan in this experiment? Also, it is recommended to change WT to SPF in the figure and text, as no genetic manipulation was involved in this figure.

Thank you for your valuable feedback and insightful suggestions. In our study, we tried to identify bacteria from murine gut contents and feces using 16S sequencing. However, only *E. coli* and *Staphylococcus sciuri* were identified (Figure 1D). Consequently, our experiments were limited to these two bacterial strains. While we have not tested other bacteria, our data suggest that not all bacteria can induce the expression of Chi3l1. Given that *E. coli* is Gram-negative and *Staphylococcus sciuri* is Gram-positive, we hypothesized that the difference in their ability to induce Chi3l1 expression might be due to variations between Gram-negative and Gram-positive bacteria, such as the presence of lipopolysaccharides (LPS).

To test this hypothesis, we used LPS to induce Chi3l1 expression. Consistent with our hypothesis, LPS successfully induced Chi3l1 expression (Figure 1F&G). Additionally, we observed that Chi3l1 expression is significantly upregulated in specific pathogen-free (SPF) mice compared to germ-free mice (Figure 1A), demonstrating that the gut microbiome induces the expression of Chi3l1.

Although we have not examined cell death or inflammatory responses, the protective role of Chi3l1 shown in Figure 5 suggests that any such responses would be mild and negligible. Regarding the role of peptidoglycan in the induction of Chi3l1 expression in DLD-1 cells, we have not yet explored this aspect. However, we agree with your suggestion that it would be worthwhile to investigate this in future experiments.

We have also made the suggested modifications to the labeling (Figure 1A) and the clarification in the revised manuscript accordingly (page 3, Line 95-96; Line 102-106).

Thank you again for your constructive feedback.

(2) In Figure 2, the binding between Chi3l1 and PGN needs better characterization, regarding the affinity and how it compares with the binding between Chi3l1 and chitin. More importantly, it is unclear how this interaction could facilitate the colonization of gram-positive bacteria.

Thank you for your insightful suggestions and we have performed the suggested experiments and included the results in the revised manuscript (Figure 2E-G, page 3-4, Line 132-146).

Our results indicate that Chi3l1 interact with PGN in a dose-increase manner (Figure 2E). In contrast, the binding between Chi3l1 and chitin did not exhibit dose dependency (Figure 2E). These findings suggest a specific and distinct binding mechanism for Chi3l1 with PGN compared to chitin.

We conducted DLD-1 cell-bacteria adhesion experiments, using GlmM mutant (PGN synthesis mutant) and K12 (wild-type) bacteria to test their adhesion capabilities. The results showed that the adhesion ability of the GlmM mutant to cells significantly decreased (Figure 2F). Additionally, after knocking down Chi3l1 in DLD-1 cells, we observed a decreased bacterial adhesion (Figure 2G). These findings suggest that Chi3l1 and PGN interaction plays a crucial role in bacterial adhesion.

(3) In Figure 3, the abundance of furmicutes and other gram-positive species is lower in the knockout mice. What is the rationale for choosing lactobacillus in the following transfer experiments?

We appreciate your thorough review. Among the Gram-positive bacteria that we have sequenced and analyzed, *Lactobacillus* occupies the largest proportion. Given the significant presence and established benefits of *Lactobacillus*, we chose it for the subsequent transfer experiments to leverage its known properties and availability, thereby ensuring the robustness and reproducibility of our findings.This is supported by the study referenced below.

Lamas B, Richard ML, Leducq V, Pham HP, Michel ML, Da Costa G, Bridonneau C, Jegou S, Hoffmann TW, Natividad JM, Brot L, Taleb S, Couturier-Maillard A, Nion-Larmurier I, Merabtene F, Seksik P, Bourrier A, Cosnes J, Ryffel B, Beaugerie L, Launay JM, Langella P, Xavier RJ, Sokol H. CARD9 impacts colitis by altering gut microbiota metabolism of tryptophan into aryl hydrocarbon receptor ligands. Nat Med. 2016 Jun;22(6):598-605. doi: 10.1038/nm.4102. Epub 2016 May 9. PMID: 27158904; PMCID: PMC5087285.

(4) FDAA-labeled *E. faecalis* colonization is decreased in the knockouts. Is it specific for E. faecalis, or it is generally true for all gram-positive bacteria? What about the colonization of gram-negative bacteria?

Thank you for your insightful suggestions and we have investigated the colonization of gram-negative bacteria, OP50-mcherry (a strain of *E.coli* that express mCherry) and included the results in the updated manuscript (Supplementary Figure 3B, page 5, Line 197-200). We performed rectal injection of both wildtype and Chi11-/- mice with mCherry-OP50, and found that Chi11-/- mice had much higher colonization of *E. coli* compared to wildtype mice.

(5) In Figure 5, the fact that FMT did not completely rescue the phenotype may point to the role of host cells in the processes. The reason that lactobacillus transfer did completely rescue the phenotypes could be due to the overwhelming protective role of lactobacillus itself, as the experiments were missing villin-cre mice transferred with lactobacillus.

Thank you for your valuable feedback and thorough review. In our study, pretreatment with antibiotics in mice to eliminate gut microbiota demonstrated that IEC∆Chil1 mice exhibited a milder colitis phenotype (Supplementary Figure 4). This suggests that Chi3l1-expressing host cells are likely to play a detrimental role in colitis. Consequently, the failure of FMT to completely rescue the phenotype is likely due to the incomplete preservation of bacteria in the feces during the transfer experiment.

We agree with your assessment of the protective role of *lactobacillus*. This also explains the significant difference in colitis phenotype between Villin-cre and IEC∆Chil1 mice (Figure 5B-E), as *lactobacillus* levels are significantly lower in IEC∆Chil1 mice (Figure 4F). Given the severity of colitis in Villin-cre mice at 7 days post-DSS, even if *lactobacillus* were transferred back to these mice, it is unlikely to result in a significant improvement.

(6) Conflicting literature demonstrating the detrimental roles of Chi3l1 in mouse IBD model needs to be acknowledged and discussed.

Thank you for your insightful suggestions and we have included additional discussions in the revised manuscript (page 6-7, Line 258-274).

**Reviewer #2 (Public Review):**
(1) Images are of great quality but lack proper quantification and statistical analysis. Statements such as "substantial increase of Chi3l1 expression in SPF mice" (Fig.1A), "reduced levels of Firmicutes in the colon lumen of IEC ∆ Chil1" (Fig.3F), "Chil1-/- had much lower colonization of *E. faecalis*" (Fig.4G), or "deletion of Chi3l1 significantly reduced mucus layer thickness" (Supplemental Figure 3A-B) are subjective. Since many conclusions were based on imaging data, the authors must provide reliable measures for comparison between conditions, as long as possible, such as fluorescence intensity, area, density, etc, as well as plots and statistical analysis.

Thank you for your insightful suggestions and we have performed the suggested statistical analysis on most of the figures and included the analysis in the revised manuscript (Figure 1A, Figure 3E&F, Supplementary Figure 3B&C).Given large quantity of dietary fiber intertwined with bacteria, it is challenging to make a reliable quantification of bacteria in Figure 4G. However, it is easy to distinguish bacteria from dietary fiber under the microscope. We have exclusively analyzed gut sections from six mice in each group, and the results are consistent between the two groups.

(2) In the fecal/Lactobacillus transplantation experiments, oral gavage of Lactobacillus to IECΔChil1 mice ameliorated the colitis phenotype, by preventing colon length reduction, weight loss, and colon inflammation. These findings seem to go against the notion that Chi3l1 is necessary for the colonization of Lactobacillus in the intestinal mucosa. The authors could speculate on how Lactobacillus administration is still beneficial in the absence of Chi3l1. Perhaps, additional data showing the localization of the orally administered bacteria in the gut of Chi3l1 deficient mice would clarify whether Lactobacillus are more successfully colonizing other regions of the gut, but not the mucus layer. Alternatively, later time points of 2% DSS challenge, after Lactobacillus transplantation, would suggest whether the gut colonization by Lactobacillus and therefore the milder colitis phenotype, is sustained for longer periods in the absence of Chi3l1.

Thank you for your thorough review and insightful suggestions. Since we pretreated mice with antibiotics, the intestinal mucus layer is likely damaged according to a previous study (PMID: 37097253). Therefore, gavaged *Lactobacillus* cannot colonize in the mucus layer. Moreover, existing studies have shown that the protective effect of *Lactobacillus* is mainly derived from its metabolites or thallus components, rather than the living bacteria itself (PMID: 36419205, PMID: 27516254).

Zhan M, Liang X, Chen J, Yang X, Han Y, Zhao C, Xiao J, Cao Y, Xiao H, Song M. Dietary 5-demethylnobiletin prevents antibiotic-associated dysbiosis of gut microbiota and damage to the colonic barrier. Food Funct. 2023 May 11;14(9):4414-4429. doi: 10.1039/d3fo00516j. PMID: 37097253.

Montgomery TL, Eckstrom K, Lile KH, Caldwell S, Heney ER, Lahue KG, D'Alessandro A, Wargo MJ, Krementsov DN. Lactobacillus reuteri tryptophan metabolism promotes host susceptibility to CNS autoimmunity. Microbiome. 2022 Nov 23;10(1):198. doi: 10.1186/s40168-022-01408-7. PMID: 36419205.

Piermaría J, Bengoechea C, Abraham AG, Guerrero A. Shear and extensional properties of kefiran. Carbohydr Polym. 2016 Nov 5;152:97-104. doi: 10.1016/j.carbpol.2016.06.067. Epub 2016 Jun 23. PMID: 27516254.

**Reviewer #3 (Public Review):**
The claim that mucus-associated Ch3l1 controls colonization of beneficial Gram-positive species within the mucus is not conclusive. The study should take into account recent discoveries on the nature of mucus in the colon, namely its mobile fecal association and complex structure based on two distinct mucus barrier layers coming from proximal and distal parts of the colon (PMID:). This impacts the interpretation of how and where Ch3l1 is expressed and gets into the mucus to promote colonization. It also impacts their conclusions because the authors compare fecal vs. tissue mucus, but most of the mucus would be attached to the feces. Of the mucus that was claimed to be isolated from the WT and IEC Ch3l1 KO, this was not biochemically verified. Such verification (e.g. through Western blot) would increase confidence in the data presented. Further, the study relies upon relative microbial profiling, which can mask absolute numbers, making the claim of reduced overall Gram-positive species in mice lacking Ch3l1 unproven. It would be beneficial to show more quantitative approaches (e.g. Quantitative Microbial Profiling, QMP) to provide more definitive conclusions on the impact of Ch3l1 loss on Gram+ microbes.

You raise an excellent point about the data interpretation, and we appreciate your insightful suggestions. We have included the discussion regarding the recent discoveries in the revised manuscript (page 7-8, Line 304-312). According to the recent discovery, the mucus in the proximal colon forms a primary encapsulation barrier around fecal material, while the mucus in the distal colon forms a secondary barrier. Our findings indicate that Chi3l1 is expressed throughout the entire colon, including the proximal, middle, and distal sections (See Author response image 1 below, P.S. Chi3l1 detection in colon presented in the manuscript are from the middle section). This suggests that Chi3l1 likely promotes bacterial colonization across the entire colon. Despite most mucus being expelled with feces, the

constant production of mucus and the minimal presence of Chi3l1 in feces (Figure 4C) indicate that Chi3l1 continuously plays a role in promoting the colonization of microbiota.

**Author response image 1. sa3fig1:** Chi3l1 express in the proximal and distal colon. Immunofluoresence staining on proximal and distal colon sections to detect Chi3l1 (Red) expression. Nuclei were detected with DAPI (blue). Scale bars, 50 μm.

Given the isolation method of the mucus layer, we followed the paper titled "The Antibacterial Lectin RegIIIγ Promotes the Spatial Segregation of Microbiota and Host in the Intestine" (PMID: 21998396). Although we did not find a suitable marker representative of the mucus layer for western blotting, we performed protein mass spectrometry on the isolated mucus layers and analyzed the data by comparing it with established research ("Proteomic Analyses of the Two Mucus Layers of the Colon Barrier Reveal That Their Main Component, the Muc2 Mucin, Is Strongly Bound to the Fcgbp Protein," PMID: 19432394). Our data showed a high degree of overlap with the proteins identified in established studies (see Author response image 2 below).

**Author response image 2. sa3fig2:** Comparison of mucus layer proteins identified by mass spectrometry between our team and the Hansson team. Mucus layer proteins identified by mass spectrometry between our team and the Hansson team (PMID: 19432394) are compared.

Due to a lack of expertise, it has been challenging for us to perform reliable QMP experiments. However, since QMP involves qPCR combined with bacterial sequencing, we conducted 16S rRNA sequencing and confirmed the quantity of certain bacteria by qPCR (revised manuscript, Figure 3B, H, Figure 4E, F, Supplementary Figure 3A). Therefore, our data is reliable to some extent.

Other weaknesses lie in the execution of the aims, leaving many claims incompletely substantiated. For example, much of the imaging data is challenging for the reader to interpret due to it being unfocused, too low of magnification, not including the correct control, and not comparing the same regions of tissues among different in vivo study groups. Statistical rigor could be better demonstrated, particularly when making claims based on imaging data. These are often presented as single images without any statistics (i.e. analysis of multiple images and biological replicates). These images include the LTA signal differences, FISH images, Enterococcus colonization, and mucus thickness.

Thank you for your thorough review and insightful suggestions. We have performed the recommended statistical analysis on most of the figures and included the analysis in the revised manuscript (Figure 1A, Figure 3E&F, Supplementary Figure 3B&C). We have also added arrows in Figure 2B to make the figure easier to understand. Additionally, we repeated some key experiments to show the same regions of tissues among different groups. We will upload higher resolution figures during the revision. Thank you again for your constructive feedback.

**Recommendations for the authors:**

**Reviewer #1 (Recommendations For The Authors):**
It is recommended to change WT to SPF in the figure and text, as no genetic manipulation was involved in Figure 1.

Thank you for your insightful suggestion. We have also made the suggested modifications to the labeling (revised manuscript, Figure 1A).

**Reviewer #2 (Recommendations For The Authors):**
The manuscript is well-written, but it would benefit from a critical reading to correct some typos and small grammar issues. Histological and IF images would be more informative if they contained arrows and labels guiding the reader's attention to what the authors want to show. More details about the structures shown in the figures should be included in the legends.

Thank you for your thorough review and insightful suggestions. We have revised the manuscript to correct noticeable typos and grammar issues. Arrows have been added to Figure 2A&B to make the figures easier to understand. Additionally, we have included a detailed description of the structural similarities and differences between chitin and peptidoglycan in the figure legend (revised manuscript, page 19, line 730-733).

Minor points:• Page 1, line 36: Please correct "mice models" to "mouse models".

Thank you for your insightful suggestion and we have made the suggested correction in the revised manuscript (page 1, line 41).

• Page 3, line 110: "by comparing the structure of chitin with that of peptidoglycan (PGN), a component of bacterial cells walls, we observed that they have similar structures (Fig.2A)". Although both structures are shown side-by-side, no similarities are mentioned or highlighted in the text, figure, or legend.

Thank you for your insightful suggestion and we have included a detailed description of the structural similarities and differences between chitin and peptidoglycan in the figure legend (revised manuscript, page 19, line 730-733).

• Fig.5C and Fig.5G: y axis brings "weight (%)". I believe the authors mean "weight change (%)"?

We agrees with your suggestion and has corrected the labeling according to your suggestion (revised manuscript, Figure 5C and G)

• Page 8: Genotyping method is described as a protocol. Please modify it.

Thank you for your constructive suggestion and we have modified the genotyping method in the revised manuscript (page 8, line 339-349)

• Please expand on the term "scaffold model" used in the abstract and discussion.

Thank you for your thorough review. In this model, Chi3l1 acts as a key component of the scaffold. By binding to bacterial cell wall components like peptidoglycan, Chi3l1 helps anchor and organize bacteria within the mucus layer. This interaction facilitates the colonization of beneficial bacteria such as Lactobacillus, which are important for gut health. We included more descriptions regarding scaffold model in the revised manuscript (page 6, line 248-250)

• Discussion session often recapitulates results description, which makes the text repetitive.

Thank you for your constructive suggestion and we have removed unnecessary results description in the discussion session in the revised manuscript.

**Reviewer #3 (Recommendations For The Authors):**
Major comments(1) Figure 1A. The staining is very faint, and hard to see. The reader cannot be certain those are Ch311-positive cells. Higher Mag is needed.

Thank you for your insightful suggestion and we have included the higher resolution figures in the revised manuscript Figure 1A.

(2) The mucus is produced largely by the proximal colon, is adherent to the feces, and mobile with the feces (PMID: 33093110). Therefore it is important to determine where the Ch311 is being expressed to be released into the lumen. Further Ch3l1 expression studies are needed to be done in both proximal and distal colon.

Thank you for your thorough review and insightful suggestions. We have addressed this part in our public review. Additionally, we agree with your suggestions and will conduct further studies on Chi3l1 expression in both the proximal and distal colon.

(3) Figure 1B. The image is out of focus for the Ileum, and the DAPI signal needs to be brought up for the colon. Which part of the colon is this? The UEA1+ cells do not really look like goblet cells. A better image with clearer goblet cells is needed.

Thank you for your constructive suggestions. In the revised manuscript, we have included higher-resolution images (Figure 1B). The middle colon (approximately 3 to 4 cm distal from the cecum) was harvested for staining. In addition to UEA-1, we utilized anti-MUC2 antibody to label goblet cells in this colon segment (see Author response image 3 below). The patterns of goblet cells identified by UEA-1 or MUC2 antibodies are similar. The UEA-1-positive cells shown in Figure 1B are presumed to be goblet cells.

**Author response image 3. sa3fig3:** Goblet Cell Distribution in the Middle Colon. Goblet cells in the middle segment of the colon (approximately 3 to 4 cm distal from the cecum) were detected using immunofluorescence with antibodies against UEA-1 (green) and MUC2 (red). Scale bars = 50 μm. Representative images are shown from three mice individually stained for each antibody.

(4) Figure 1G. There needs to be some counterstain or contrast imaging to show evidence that cells are present in the untreated sample.

Thank you for your insightful suggestions. We have annotated the cells present in the untreated sample based on the overexposure in the revised manuscript (Figure 1G).

(5) Figure 3B. Is this absolute quantification? How were the data normalized to allow comparison of microbial loads?

Thank you for your thorough review. Figure 3B presents absolute quantification data based on the methodology described in the paper titled "The Antibacterial Lectin RegIIIγ Promotes the Spatial Segregation of Microbiota and Host in the Intestine" (PMID: 21998396). Briefly, we amplified a short segment (179 bp) of the 16S rRNA gene using conserved 16S rRNA-specific primers and OP50 (a strain of *E. coli*) as the template. After gel extraction and concentration measurement, the PCR products were diluted to gradient concentrations (0.16, 0.32, 0.64, 1.28, 2.56, 5.12, 10.24, 20.48 pg/µl). These gradient concentrations were used as templates for qPCR to generate a standard curve based on Ct values and bacterial concentration. The standard curve is used to calculate bacterial concentration in the samples. The data presented in Figure 3B represent the weight of bacteria/milligram sample, calculated as (bacterial concentration x bacterial volume) / (weight of feces or gut content).

(6) Figure 3D. The major case is made for a dramatic reduction in Gram+ species, but Figure 1D does not show a dramatic change. Is this difference significant?

Thank you for your thorough review. We don’t think we are clear about your question. However, there was no significant difference in Figure 3D. The dramatic reduction in Gram+ species are made based on the LTA, Firmicutes FISH, individual species comparison between WT and KO mice, bacterial QPCR results together (Figure 3E-H).

(7) Figures 3E and 3F. These stainings are alone not convincing of reduced Gram+ in the KOs. Some stats are required for these images. An independent complementary method is also needed to quantify these with statistics since this data is so central to the study's conclusions.

Thank you for your constructive suggestions. We have included statistical analysis in the revised manuscript (Figure 3E and F). Given large quantity of dietary fiber intertwined with bacteria, it is challenging to make a reliable quantification of bacteria in Figure 3E. However, it is easy to distinguish bacteria from dietary fiber under the microscope. We have exclusively analyzed gut sections from six mice in each group, and the results are consistent with the Firmicutes FISH results. Complementary method such as bacterial QPCR have been employed to quantify these (Figure 4E, F). Due to a lack of expertise, it has been challenging for us to perform reliable QMP experiments.

(8) Figure 3G. To make quantitative conclusions, the authors need to do quantitative microbial profiling (QMP) of the microbiota. Relative abundance masks absolute numbers, which could be increased. There are qPCR-based QMP platforms the authors could use (PMID: PMIDs: 31940382, 33763385).

Thank you for your constructive suggestions. Due to a lack of expertise, it has been challenging for us to perform reliable QMP experiments. However, since QMP involves qPCR combined with bacterial sequencing, we conducted 16S rRNA sequencing and confirmed the quantity of certain bacteria by qPCR (revised manuscript, Figure 3B, H, Figure 4E, F, Supplementary Figure 3A). In addition to the original bacterial qPCR data presented in the manuscript, we included another bacterial species, *Turicibater.* Consistent with the 16S rRNA sequencing analysis data, qPCR results showed that Turicibacter was more abundant in IECΔChil1 mice than Villin-cre mice (revised manuscript, supplementary Figure 3A, page 4, line 171-173) Therefore, our data is reliable to some extent.

(9) Figure 4B. The data nicely shows Ch3l1 in mucus. However, no data supports the authors' main claim Ch3h1 binds Gram-positive bacteria in situ. Dual staining of Ch3l1 with Firmicutes probe would be supportive to show this interaction is happening in vivo.

You raise an excellent point, and we agree with your suggestion that we should confirm Chi3l1 binding to Gram-positive bacteria in situ. During the study, we attempted dual staining of Chi3l1 with a universal bacterial 16S FISH probe several times, but we were unsuccessful. Despite various optimizations of the protocol, we were only able to detect bacteria, not Chi3l1. It appears that the antibody is not suitable for this method.

(10) Figures 4D - F. Because mucus is associated with feces (PMID:), the data with feces likely contains both Muc2/mucus and Feces. Therefore, it is unclear what the "mucus" is referring to in these figures. To support the authors' conclusions, there needs to be some validation that mucus was purified in the assays. This must be confirmed at a minimum by PAS staining on SDS PAGE gel (should be very high molecular weight) or Western blot with UEA lectin.

Thank you for your insightful suggestions. As mentioned in the public review, the mucus layer was isolated following the protocol described in the paper titled "The Antibacterial Lectin RegIIIγ Promotes the Spatial Segregation of Microbiota and Host in the Intestine" (PMID: 21998396). Briefly, after harvesting the middle colon from the mice, we cut open the colon longitudinally. After removing the gut contents, the lumen was vigorously rinsed in PBS while holding one end with forceps. The pellet obtained after centrifuging the rinsate was used as our mucus sample. Fresh feces were collected immediately after the mice defecated in a new, empty cage. We performed Western blot analysis to detect UEA lectin but were unsuccessful.

However, as noted in the public review, we conducted protein mass spectrometry on the isolated mucus layers and analyzed the data by comparing it with established research ("Proteomic Analyses of the Two Mucus Layers of the Colon Barrier Reveal That Their Main Component, the Muc2 Mucin, Is Strongly Bound to the Fcgbp Protein," PMID: 19432394). Our data showed a high degree of overlap with the proteins identified in these established studies.

(11) Figure 4E/F: The units of measurement are in pg/cm2, implying picogram per area. Can the authors please explain what this unit is referring to?

We are grateful for your thorough review. The unit pg/cm ² represents picograms per square centimeter. Figures 4E and 4F present absolute quantification data based on the methodology described in the paper titled "The Antibacterial Lectin RegIIIγ Promotes the Spatial Segregation of Microbiota and Host in the Intestine" (PMID: 21998396). Briefly, we harvested a 3x0.5 cm section of colon and a 9x0.4 cm section of ileum. And then we collected the mucus layer as previously described (responses to question 10). We measured bacterial concentration as described in response to question 5 using the equation (y = -1.53ln(x) + 13.581), where x represents the bacterial concentration and y represents the Ct value. After obtaining the bacterial concentration, we multiplied it by the volume of the rinsate and divided it by the area to obtain the values for pg/cm² used in the figures.

(12) Figure 5E. Normal tissues appear to be from different colon regions from colitis tissues: the "Normal" looks like the proximal colon, while "Colitis" looks like the Distal colon. They cannot be directly compared.

Thank you for your insightful suggestion. We have now included the updated image in the revised manuscript as Figure 5E to compare the same region of the colons.

(13) Similarly, in Figure 5I it appears different colon regions are being compared between groups: Proximal colon in the bottom panels, and distal in the top panels. Since the proximal colon is less damaged by DSS, this data could be misleading.

Thank you for your insightful suggestion. We have now included the updated image in the revised manuscript as Figure 5I to compare the same region of the colons.

(14) In the DSS studies, are the VillinCre and IEC Chit3l1 mice co-housed littermates?

Thank you for your insightful suggestion. In the DSS studies, the Villin-Cre and IECΔChil1 mice are not co-housed littermates. However, they are derived from the same lineage and are housed in the same rack within the same room of the animal facility.

(15) Supplementary Figure 3: Mucus thickness images; are they representative? Stats are needed on multiple mice to support the claim that the mucus is thinner.

Thank you for your insightful suggestion. The images are representative of 4 mice each group. We have now included the statistical analysis in the revised manuscript Supplementary Figure 3C&D.

Minor(1) Introduction: Reference to "mucosal layer": "Mucosal" and "Mucus" are different things. "Mucosal" refers to the epithelium, lamina propria, and muscularis mucosa. "Mucus" refers to the secreted mucus gel, the focus of the authors' study. Therefore, the statement "mucosal layer" is not proper. "Mucosal layer" should be changed to "mucus layer."

Thank you for your constructive suggestions and we have learned a lot from it. We have made the replacement of “mucosal layer” to “mucus layer in the revised manuscript.

(2) Line 366 and related lines: Feces cannot be "dissolved". "Resuspended" is a better term.

Thank you for your constructive suggestion and we have made the changes of “dissolved” to “resuspended” in the revised manuscript.

(3) Lines 36-37 and 43-44 are redundant to each other.

Thank you for your constructive suggestion and we have removed the lines 36-37 in the revised manuscript.